



# Regional evaluation of the performance of the global CAMS chemical modeling system over the United States (IFS cycle 47r1)

Jason E. Williams[1], Vincent Huijnen[1], Idir Bouarar[2], Mehdi Meziane[3], Timo Schreurs[4], Sophie Pelletier[3], Virginie Marécal[3] Beatrice Josse[3] and Johannes Flemming[4]

1 R&D Weather and Climate Modeling, Royal Netherlands Meteorological Institute, De Bilt, the Netherlands

2 Max Planck Institute for Meteorology, Hamburg, Germany

3 Centre National de Recherches Météorologiques, Université de Toulouse, Météo-France, CNRS, Toulouse, France

4 ECMWF, Shinfield Park, Reading, UK

Correspondance to: Jason. E Williams (Jason.Williams@knmi.nl)

**Abstract** The Copernicus Atmosphere Monitoring Service (CAMS) provides routine analyses and forecasts of trace gases and aerosols on a global scale. The core is ECMWF's Integrated Forecast System (IFS), where modules for atmospheric chemistry and aerosols have been introduced, and which allows data-assimilation of satellite retrievals of composition.

We have updated both the homogeneous and heterogeneous $NO_x$ chemistry applied in the three independent tropospheric-stratospheric chemistry modules maintained within CAMS, referred to as IFS(CB05BASCOE), IFS(MOCAGE) and IFS(MOZART). Here we focus on the evaluation of main trace gas products from these modules that are of interest as markers of air quality, namely lower tropospheric $O_3$, $NO_2$ and CO, with a regional focus over the contiguous United States without data assimilation.

Evaluation against lower tropospheric composition reveals overall good performance, with chemically induced biases within 10 ppb across species across regions within the US with respect to a range of observations. The versions show overall equal or better performance than the CAMS Reanalysis. Evaluation of surface air quality aspects shows that annual cycles are captured well, albeit with variable seasonal biases. During wintertime conditions there is a large model spread between chemistry schemes in lower-tropospheric $O_3$ (~10-35%) and, in turn, oxidative capacity related to $NO_x$ lifetime differences. Analysis of differences in the $HNO_3$ and PAN formation, which act as reservoirs for reactive nitrogen, revealed a general underestimate in PAN formation over polluted regions likely due to too low organic precursors. Particularly during wintertime, the fraction of $NO_2$ sequestered into PAN has a variability of 100% across chemistry modules indicating the need for further constraints. Notably a considerable uncertainty in $HNO_3$ formation associated with wintertime $N_2O_5$ conversion on wet particle surfaces remains.

In summary this study has indicated that the chemically induced differences in the quality of CAMS forecast products over the United States depends on season, trace gas, altitude and region. Whilst analysis of the three chemistry modules in CAMS provide a strong handle on uncertainties associated with chemistry modeling, the further improvement of operational products additionally requires coordinated development involving emissions handling, chemistry and aerosol modeling, complemented with data-assimilation efforts.




## 1 Introduction

Poor air quality has a significant impact on visibility, human health and lifespan, crop production and ecosystems, while this impact is expected to be accentuated due to climatic change (Silva et al., 2017; Reddington et al., 2019;
Schneidemesser et al., 2020). High concentrations of pollutants induce premature mortality (e.g. Lelieveld et al., 2015) and spark episodes in people with asthma. For these reasons a predictive capability at local scale is deemed desirable in order to provide forewarning of intense pollution episodes and to perform retrospective monitoring of annual exposure. Thus, like for forecasting weather events, during the last decades there has been a focus on integrating interactive chemistry and aerosol modules into global weather-forecasting models such as the European
Centre for Medium Range Weather Forecasts (ECMWF) Integrated Forecasting System (IFS) (Benedetti et al., 2009; Morcrette et al., 2009; Flemming et al., 2015; Huijnen et al., 2016; 2019; Rémy et al., 2019) amongst others for the purpose of providing short-term Air-Quality Forecasts (AQF) at global scale, in the framework of the Copernicus Atmosphere Monitoring Service (CAMS). The IFS system is upgraded frequently, allowing the fast benefit of advances in meteorological aspects, the chemistry modeling and its interactions. Note that beside the
global system, an operational suite of regional-scale models for providing timely AQF for Europe (Marécal et al., 2015). For other domains such as the United States (US), AQF are provided by the global CAMS system (http://atmosphere.copernicus.eu; last access 28.09.21), updated twice daily and run at a resolution of approximately 0.4° x 0.4° on 137 vertical levels. The operational configuration furthermore relies on data assimilation of trace gases and aerosol for a suite of satellite retrievals, combined with a state-of-the-art
atmospheric chemistry and aerosol model. Until recently, model development of the CAMS global system has focused on the performance at global scale with emphasis on more pristine regions and conditions (Huijnen et al., 2019). Only limited attention has been made for regions directly affected by high anthropogenic emission sources such as the US for which AQF are provided. This gives motivation for the study presented here, which assesses the quality of the CAMS global chemistry modeling with regard to the seasonal mean performance in tropospheric
pollutants for a typical year, as compared against a suite of independent measurements for the US.

For the operational suite, the CAMS system adopts the IFS(CB05) version of the model (i.e., based on tropospheric chemistry originating from the TM5 chemistry transport model (Huijnen et al., 2010; Williams et al., 2013) without the explicit modelling of the stratospheric ozone chemistry). This allows for convenience with respect to stability and run time, recognizing thus far the focus of CAMS products is mainly on the troposphere, while stratospheric
ozone could sufficiently well be constrained by a linear model combined with (ozone) data assimilation, e.g. Inness et al. (2020). Another application of this system has been the production of a consistent, long-term reanalysis dataset from 2003 to present (Inness et al., 2019; Wagner et al., 2021), which can be used to analyze interannual variability in atmospheric composition (Huijnen et al., 2020). Also, as the reanalysis is using a fixed model configuration (IFS cycle 42R1), this dataset can be as well used as a reference for assessing changes in the
performance of the IFS and updates to the operational chemistry modules since that cycle. One constraint is that whatever changes are adopted there should be a net improvement of key products, such as tropospheric $O_3$, as measured with respect to a comprehensive set of reference observations. Key trace gases from the reanalysis dataset (hereafter referred to as CAMSRA) have recently been compared against a host of different aircraft data to allow an assessment of biases at global scale over multiple years (Wang et al., 2020). For CO and $O_3$, regional
negative biases of between 15-30% were found, where large biases have also been found in both OH and HO2 radicals which act as key oxidants in the troposphere. For CO there is typically an underestimation in the simulated surface concentration (Huijnen et al., 2019), albeit for regions far away from high emission sources, while this is efficiently corrected by the data assimilation in CAMSRA. For the surface over the North Hemispheric mid-latitude continents, CAMSRA typically shows seasonal biases in the monthly mean $O_3$, with a negative bias during
wintertime and a positive bias during summertime (Wagner et al., 2021). A similar performance is noticed for its control simulation which excludes data assimilation of trace gases. This indicates that further improvements can be attained by focusing on improving the relevant chemical processes included in the CAMS operational system because the impact of the data assimilation of atmospheric composition is limited at the model surface and does not constrain many key species.

To date within the CAMS global modelling system three independent atmospheric chemistry modules are maintained apart from the operational configuration, as described in Huijnen et al. (2019). These are the modules based on the modified CB05 scheme (Williams et al., 2013; 2017), optionally coupled to the BASCOE





stratospheric chemistry as described in Huijnen et al. (2016); the RACM/REPROBUS module originating from MOCAGE (Lacressonnière et al., 2012, Cussac et al., 2020) and the MOZART module (Kinnison et al., 2007, Emmons et al., 2010). One main difference is that the CB05 uses a reactive group lumping approach for a selection of Volatile Organic Compounds (VOC's), whereas the other schemes describe the degradation of organic compounds more explicitly for e.g. aromatic organics. Other important differences concern the parameterization of photolysis rates and details associated with the parameterization of inorganic chemistry. Huijnen et al. (2019) has highlighted the importance of heterogeneous reactions to explain differences. When applied in the IFS, excluding data assimilation, these mechanisms have been shown to lead to differences of ~10% for tropospheric ozone ($O_3$) and ~20% for tropospheric carbon monoxide (CO), due to the diverse treatment of Volatile Organic Compounds (VOCs) and variability in oxidative capacity via differences in the hydroxyl radical (OH), Huijnen et al. (2019). To date global validation has been focused on background conditions, with assessment of seasonal averages on a continental scale. Furthermore, by applying all three model versions in the form of a chemistry mini-ensemble, uncertainty ranges due to inaccuracies in the chemical component of the forecast can be estimated. Such information could be provided operationally on the condition that the runtimes of forecasts for each member are similar, and that such a model spread is meaningful.

Tropospheric $O_3$ is principally formed via the photolysis of nitrogen dioxide ($NO_2$), with the regeneration of $NO_2$ occurring via the oxidation of nitric oxide (NO) with peroxy-radicals ($HO_2$, $CH_3O_2$, $RO_2$) and the titration of $O_3$. The chain length of this cycle is determined by the loss of $NO_2$ into more stable nitrogen compounds, namely nitric acid ($HNO_3$), peroxy-acetyl nitrate (PAN) and organic nitrates (commonly referred to as $NO_y$ species), where a large fraction of $HNO_3$ is formed via the heterogeneous conversion of nitrogen pentoxide ($N_2O_5$) on wet surfaces (e.g. Brown et al., 2009). $HNO_3$ is soluble and is lost via wet deposition and/or the formation of inorganic aerosol particles in the form of nitrate ($NO_3^-$), whereas PAN exists in a thermal equilibrium. It can dissociate to release $NO_2$ allowing transport of reactive $NO_x$ away from the source regions (Fischer et al., 2014). The length of the $NO_x$ cycle depends on both the chemical mechanism and the rate parameters employed, determining the regional $O_3$ production efficiency. Therefore, to fully understand differences in $O_3$ production efficiency between the various chemical schemes requires analysis of the major $NO_y$ components.

In this study we present an evaluation of key trace gas products (tropospheric $O_3$, $NO_2$, CO) simulated by the chemistry modules implemented in the CAMS system for the contiguous United States. This evaluation is performed for the years 2014-2015, spanning an entire summer and winter period. We also include the corresponding products from the CAMS Reanalysis dataset (Inness et al., 2019) to provide an anchor point towards this previous model version. In Sect. 2 we provide details of the chemistry modules employed in the CAMS system, with emphasis on recent updates to all three chemical mechanisms. In Sect. 3 we provide details of the observations used for evaluating the regional performance across the United States. In Sect. 4, we present analyses for the three main chosen gases and in Sect. 5 we provide further discussion concerning the variability in the main trace species across the different modules with a focus on differences due to $NO_y$. Finally in Sect. 6 we present our conclusions. Additional information in support of the main findings is also provided in the supplementary material.

## 2 Model Description

In this section we provide a brief description of the various configurations of the CAMS system for global atmospheric chemistry modelling. Here we focus on the upgrades which have been made to the three chemistry modules (CB05BASCOE, MOCAGE, MOZART) available in the IFS as compared to the extensive description of each of the modules as provided in Huijnen et al. (2019). Hereafter we refer to these model configurations as IFS(CBA), IFS(MOC) and IFS(MOZ), respectively.

A brief overview of the contents and parameterized differences for each of the various chemical modules is provided in Table 1, below. For details pertaining to the CAMSRA reanalysis dataset the reader is referred to Innes et al. (2019). There is significant variability in the number of thermal (photolytic) reactions across schemes, with IFS(MOC) and IFS(MOZ) being the most explicit (condensed). Compared to Huijnen et al. (2019), the heterogenous scavenging and conversion for $N_2O_5$ and $HO_2$ has also been homogenized across the different schemes. As in previous versions, the calculation of photolysis rates is characteristic for each scheme, where a recent inter-comparison has been conducted by Hall et al. (2018) showing differences in the key photolysis frequencies ($J$ values) of ~5%, where the percentage cloud-cover and droplet size provided by the IFS is identical





throughout. Heterogeneous conversion and scavenging are described using the approach of Chang et al. (2011), where the loss of $HO_2$ and $NO_3$ also occur as pseudo-first-order sink processes.

**Table 1**: An overview of the three chemistry modules included in the IFS for this study with main updates compared to Huijnen et al. (2019).

|  | IFS(CBA) | IFS(MOC) | IFS(MOZ) |
|---|---|---|---|
| Tropospheric chemistry | CBM | RACM | CAM4-Chem |
| Stratospheric chemistry | BASCOE | REPROBUS | MOZART3 |
| chemistry Number of Species | 120 | 123 | 132 |
| Number of thermal reactions | 219 | 333 | 291 |
| Number of photolysis rates | 72 | 56 | 78 |
| Complexity of organic chemistry | Updated $C_5H_8$ degradation | Detailed lumping approach | Explicit degradation pathways up to C10 |
| Complexity of inorganic chemistry | Extended for HONO and $CH_3O_2NO_2$ | Extended including HONO | As for CBA |
| Aerosol interaction in the troposphere | $N_2O_5$, $NO_3$ and HO2 heterogeneous reactions on aerosol and cloud AOD influences J values. | Revised heterogeneous chemistry on aerosol and cloud as in CBA | Revised heterogeneous chemistry on aerosol included as in CBA |
| Loss on clouds and ice particles | Yes | Yes | No. |
| Photolysis parameterization | Modified Band Approach (Trop) online photolysis (Strat) | Look-up Table | Look-up Table (Trop), explicit transmission function (Strat) |
| Solver | 3rd Order Rosenbrock | 3rd Order Rosenbrock | Explicit forward and implicit backward Euler |

## 2.1 IFS(CB05BASCOE)


    IFS(CB05BASCOE), or IFS(CBA) in short, is a merge of tropospheric chemistry originally based on a modified version of the CB05 mechanism (Yarwood et al., 2005), combined with stratospheric chemistry originating from BASCOE (Skachko et al., 2016). The CB05 tropospheric chemistry in the IFS, of primary relevance in this study, adopts a lumping approach for organic species by defining a separate tracer species (Williams et al., 2013;

Flemming et al., 2015).

    The modified band approach (MBA), which is adopted for the computation of tropospheric photolysis rates (Williams et al., 2012), uses 7 absorption bands across the spectral range 202−695 nm. In the MBA, the radiative transfer calculation is performed with a two-stream solver using the absorption and scattering components introduced by gases, aerosols and clouds is computed on-line for each of the predefined band intervals.

Heterogeneous reactions and photolysis rates are calculated using CAMS IFS-AER aerosol model (Rémy et al. , 2019). The reaction rates follow the recommendations given in either Burkholder et al. (2015) or the latest recommendations by IUPAC (http://iupac.pole-ether.fr; last access 21 Sept 2021).



**Table 2**: Updates to the $NO_x$ reaction chemistry in CBA as compared to Huijnen et al (2019). [1] Atkinson et al., 2004; updated according to http://iupac.pole-ether.fr (last access : 21 sept 2021), [2] JPL 2015, [3] Atkinson et al., 2006; updated according to http://iupac.pole-ether.fr (last access : 21 Sept 2021).

| Reaction | Rate expression | Reference |
|---|---|---|
| $NO + HO_2 \rightarrow NO_2 + OH$ | $3.3 \times 10^{-12} \times \exp(270/T)$ | [1] |
| $NO_3 + HO_2 \rightarrow HNO_3$ | $4.0 \times 10^{-12}$ | [1] |
| $NO_2 + OH \rightarrow HNO_3$ | $K^0 = 3.2 \times 10^{-10} \times (300/T)^{4.5}$<br>$K^\infty = 3.0 \times 10^{-11}$ | [1] |
| $OH + NO \rightarrow HONO$ | $K^0 = 7.0 \times 10^{-31} \times (300/T)^{2.4}$<br>$K^\infty = 3.6 \times 10^{-11} \times (300/T)^{-0.3}$ | [2] |
| $HONO + h\nu \rightarrow OH + NO$ | Explicit | [1] |
| $OH + HONO \rightarrow NO_2$ | $2.5 \times 10^{-12} \times \exp(260/T)$ | [2] |
| $NO_2 + CH_3OO_2 \rightarrow PAN$ | $K^0 = 9.7 \times 10^{-29} \times (300/T)^{5.6}$<br>$K^\infty = 9.3 \times 10^{-12} \times (300/T)^{1.5}$ | [2] |
| $PAN \rightarrow NO_2 + CH_3OO_2$ | $9.9 \times 10^{-29} \times \exp(14000/T)$ | [2] |
| $PAN + h\nu \rightarrow CH_3OO_2 + NO_2$ | QY | [1] |
| $PAN + h\nu \rightarrow CH_3O_2 + NO_3$ | (1-QY) | [1] |
| $CH_3O_2 + NO_2 \rightarrow CH_3O_2NO_2$ | $K^0 = 1.0 \times 10^{-30} \times (300/T)^{4.8}$<br>$K^\infty = 7.2 \times 10^{-12} \times (300/T)^{2.1}$ | [2] |
| $CH_3O_2NO_2 \rightarrow CH_3O_2 + NO_2$ | $9.5 \times 10^{-29} \times \exp(11234/T)$ | [2] |
| $NO_2 + O_3 \rightarrow NO_3$ | $1.4 \times 10^{-13} \times \exp(-2470/T)$ | [1] |
| $NO + NO_3 \rightarrow NO_2 + NO_2$ | $1.8 \times 10^{-11} \times \exp(110/T)$ | [1] |
| $CH_3O_2 + HO_2 \rightarrow CH_3OOH$ | $3.8 \times 10^{-13} \times \exp(750/T) \times R$ | [1] |
| $CH_3O_2 + HO_2 \rightarrow HCHO$ | $3.8 \times 10^{-13} \times \exp(750/T) \times (1.-R)$ | [1] |
| $O_3 + C_2H_4 \rightarrow CH_2O + 0.26\ HO_2 + 0.12\ OH + 0.43\ CO$ | $6.82 \times 10^{-14} \times \exp(-2500/T)$ | [3] |
| $NO_3 + C_5H_8 \rightarrow 0.2\ ISPD + XO_2 + 0.8\ HO_2 + 0.8\ ORGNTR + 0.8\ ALD2 + 2.4\ PAR + 0.2\ NO_2$ | $2.95 \times 10^{-12} \times \exp(465/T)$ | [3] |
| $HNO_4 + OH \rightarrow NO_2$ | $3.2 \times 10^{-13} \times \exp(690/T)$ | [1] |
| $NO_2 + HO_2 \rightarrow HO_2NO_2$ | $K0 = 1.4 \times 10^{-31} \times (300/T)^{3.1}$<br>$K\infty = 4.0 \times 10^{-12}$ | [1] |
| $HO_2NO_2 \rightarrow NO_2 + HO_2$ | $K0 = 4.1 \times 10^{-5} \times \exp(-10650/T)$<br>$K\infty = 6.0 \times 10^{15} \exp(-11170/T)$ | [1] |
| $NO_2 + NO_3 \rightarrow N_2O_5$ | $K0 = 3.6 \times 10^{-30} \times (300/T)^{4.1}$<br>$K\infty = 1.9 \times 10^{-12} \times (300/T)^{-0.2}$ | [1] |





For IFS(CBA) there have been extensive modifications to four main components of the tropospheric chemistry module, namely: (i) the inclusion of HONO and $CH_3O_2NO_2$ into the $NO_x$ reaction cycle, (ii) the replacement of the isoprene ($C_5H_8$) oxidation scheme with a hybrid from the literature, (iii) the coupling of the formation of Secondary Organic Aerosol (SOA) to oxidation products of aromatics and (iv) the inclusion of hydrogen cyanide (HCN) and acetonitrile ($CH_3CN$) from Biomass Burning (BB) sources. This tropospheric chemistry version is referenced as 'tc06f' and is further described below.

The updated rate data for $NO_x$ chemistry and $NO_y$ components are listed in Table 2 and are based on the updates tested in the chemistry transport model TM5 (Williams et al., 2017). One important update is the use of the new recommendation for the formation of $HNO_3$ (Mollner et al., 2010) that has been shown to have significant effects on the tropospheric $O_3$ burden (Søvde et al., 2011). HONO acts as an important source of OH and NO during the early morning from efficient photolytic destruction after nocturnal build-up (Stutz et al., 2004), whereas $CH_3O_2NO_2$ alters the $NO_y$ chemistry in the free troposphere (Browne et al., 2011). Additionally, updates have also been made to the reaction data for $O_3 + C_2H_4$ and $NO_3 + C_5H_8$.

**Table 3**: Updates to the $C_5H_8$ oxidation scheme in CBA as compared to Huijnen et al. (2019). [1] Stavrakou et al. (2010), [2] Lamarque et al. (2012), [3] Myriokefalitakis et al. (2020), [4] spectral absorption data from http://iupac.pole-ether.fr (last access : 21 Sept 2021), [5] Quantum Yields as for 298K.

| Reaction | Rate expression | Reference |
|---|---|---|
| OH + ISOP → 0.65 $ISOPBO_2$ + 0.35 $ISOPDO_2$ | $2.7 \times 10^{-12}$ x exp(390/T) | [1] |
| $ISOPBO_2$ → HPALD1 + $HO_2$ | $4.1 \times 10^8$ x exp(-7700/T) | [1],[3] |
| $ISOPBO_2$ → ISPD + $CH_2O$ + $HO_2$ | $2.08 \times 10^{11}$ x exp(-8993/T) | [1] |
| $ISOPDO_2$ → HPALD2 + $HO_2$ | $4.1 \times 10^8$ x exp(-7700/T) | [1],[3] |
| $ISOPDO_2$ → ISPD + $CH_2O$ + $HO_2$ | $2.08 \times 10^{11}$ x exp(-8993/T) | [1] |
| $ISOPBO_2$ + $HO_2$ → ISOPOOH | $2.05 \times 10^{-13}$ x exp(1300/T) | [2] |
| $ISOPBO_2$ + NO → 0.08 ORGNTR + 0.92 $NO_2$ + $HO_2$ + 0.51 $CH_2O$ + 0.55 ISPD + 0.37 HPALD1 | $4.4 \times 10^{-12}$ x exp(180/T) | [2] |
| $ISOPDO_2$ + $HO_2$ → ISOPOOH | $2.05 \times 10^{-13}$ x exp(1300/T) | [2] |
| $ISOPDO_2$ + NO → 0.08 ORGNTR + 0.92 $NO_2$ + $HO_2$ + 0.51 HCHO + 0.55 ISPD + 0.37 HPALD2 | $4.4 \times 10^{-12}$ x exp(180/T) | [2] |
| OH + ISOPOOH → 0.1 XO2 + 0.4 $CH_3COCHO$ + 0.3 CHOCHO + 0.12 ISOPBO2 + 0.08 ISOPDO2 | $1.52 \times 10^{-11}$ x exp(200/T) | [1] |
| ISOPOOH + h$\nu$ → 0.69 ISPD + 0.69 HCHO + $HO_2$ | Explicit | [4] |
| OH + HPALD1 → 0.65 $XO_2$ + 0.25 CHOCHO + 0.1 $CH_3COCHO$ | $1.86 \times 10^{-11}$ x exp(175/T) | [2] |
| HPALD1 + h$\nu$ → OH + $HO_2$ + 0.5 HYAC + 0.5 $CH_3COCHO$ + 0.5 GLYALD + HCHO | Explicit | [4] |
| OH + HPALD2 → 0.65 $XO_2$ + 0.25 CHOCHO + 0.1 $CH_3COCHO$ | $1.86 \times 10^{-11}$ x exp(175/T) | [1] |
| HPALD2 + h$\nu$ → $HO_2$ + OH + 0.5 HYAC + 0.5 CHOCHO + 0.5 GLYALD + HCHO | Explicit | [4] |
| OH + CHOCHO → 0.63 $HO_2$ + 1.26 CO + $C_2O_3$ | $3.1 \times 10^{-12}$ x exp(340/T) | [1] |

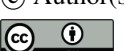



| | | |
|---|---|---|
| CHOCHO + $h\nu$ → 2 CO + 2 $HO_2$ | Explicit | [4],[5] |
| CHOCHO + $h\nu$ → HCHO + CO | Explicit | [4],[5] |
| OH + GLYALD → 0.25 OH + 0.75 $HO_2$ + 0.17 CHOCHO + 0.17 HCOOH + 0.67 HCHO + 0.5 CO | $8.0 \times 10^{-12}$ | [1] |
| GLYALD + $h\nu$ → $2HO_2$ + CO + HCHO | Explicit | [4] |
| OH + HYAC → 0.1 OH + 0.825 $HO_2$ + 0.75 $CH_3COCHO$ + 0.125 HCOOH + 0.125 $CH_3O_2$ + 0.125 $CH_3COOH$ + 0.05 CO | $2.0 \times 10^{-12} \times \exp(320/T)$ | [1] |
| HYAC + $h\nu$ → $C_2O_3$ + $HO_2$ + HCHO | As $J(CH_3COCH_3)$ | [1] |

To date IFS(CBA) has only included a very simplistic parameterization for the oxidation of $C_5H_8$. To improve the realism of the product distribution from the oxidation of $C_5H_8$ by OH, we have developed a mechanism which is a hybrid of that developed by Stavrakou et al. (2010) and Lamarque et al. (2012). This new hybrid method includes the direct formation of glyoxal (CHOCHO), hydroxy-aldehydes (HPALD1, HPALD2), a peroxy-product (ISOPOOH), glycolaldehyde (GLYALD), hydroxy-acetone (HYAC) and methyl-glyoxal ($CH_3COCHO$). Explicit $J$ values are calculated online for these products using the latest recommendations for absorption data, as for other

species. Most of these intermediates are soluble with Henry solubility analogous to ALD2 except CHOCHO, where the approach of Ip et al. (2009) is used. We still retain the ISPD intermediate representative of methyl-vinyl-ketone and methacrolein from previous versions of the scheme. Reaction rates in this mechanism have been updated following latest recommendations, as aligned with the mechanism described by Myriokefalitakis et al. (2020).

Validation of this updated component of the IFS(CBA) is beyond the scope of this study, but we have found that OH-recycling increases over forested regions with high biogenic emission fluxes is as for that shown in Stavrakou et al (2010) thus affecting atmospheric lifetime of individual trace gases for regions with high resident mixing ratios of $C_5H_8$. It should be noted that for $O_3$ and $NO_3$ we still adopt the original stoichiometry in product distribution as in previous versions of IFS(CBA) (Huijnen et al., 2016). We provide details related to this extensive

update in Table 3.

    To date aromatics were not explicitly treated in IFS(CBA), but rather as part of the generic paraffinic bond tracer PAR. This is now updated. For this we follow the work of Karl et al. (2009), who describe the oxidation of the aromatic tracers toluene (TOL) and xylene (XYL), allowing a coupling to Secondary Organic Aerosol (SOA) formation. In addition, in our model version the product distributions and rate expressions for NO and $HO_2$ radical-

radical reactions are taken from Myriokefalitakis et al. (2020). This links the aromatic species towards oxidant loss ($O_3$, OH, $NO_3$), the production of CHOCHO/$CH_3COCHO$ and allows the introduction of gas-phase precursors for SOA formation (SOG) from anthropogenic and biomass burning sources. We provide details on the extension to the aromatic chemistry in IFS(CBA) in Table 4.

**Table 4**: Updates to the oxidation of TOL and XYL as implemented in the IFS(CBA) chemistry as compared to
Huijnen et al. (2019). The reaction scheme is adapted from Karl et al. (2009), with modification to the product distribution for loss of $AROO_2$ following Myriokefalitakis et al. (2020). (*) This indicates the final rate applied accounts for the ortho-, meta- and para-isomers of the cyclic aromatics.

| Reaction | Rate expression |
|---|---|
| OH + TOL → 5PAR + $AROO_2$ | $5.96 \times 10^{-12}$ |
| $O_3$ + TOL → 5PAR + $AROO_2$ | $2.34 \times 10^{-17} \times \exp(-6694/T)$ |
| $NO_3$ + TOL → ORGNTR + PAR | $6.8 \times 10^{-17}$ |


| OH + XYL → 5PAR + AROO$_2$ | avg of ($1.3 \times 10^{-11}$, $2.36 \times 10^{-11}$, $1.43 \times 10^{-11}$)* |
|---|---|
| O$_3$ + XYL → 5PAR + AROO$_2$ | avg of ($5.37 \times 10^{-13} \times \exp(-6039/T)$, $1.91 \times 10^{-13} \times \exp(-5586/T)$, $2.4 \times 10^{-13} \times \exp(-5586/T)$) |
| NO3 + XYL → CH$_3$COCHO + PAR | avg of ($3.6 \times 10^{-16}$, $2.33 \times 10^{-16}$, $4.5 \times 10^{-16}$)* |
| NO + AROO$_2$ → NO$_2$ + CHOCHO + 0.33 CH$_3$COCHO | $4.2 \times 10^{-12} \times \exp(180/T)$ |
| XO$_2$ + AROO$_2$ | $1.7 \times 10^{-14} \times \exp(1300/T)$ |
| HO$_2$ + AROO$_2$ → ROOK + CHOCHO | $3.5 \times 10^{-13} \times \exp(1000/T)$ |
| AROO$_2$ + AROO$_2$ | $1.7 \times 10^{-14} \times \exp(1300/T)$ |

In Table 5 we provide the rate data used for the oxidation of hydrogen cyanide (HCN) and acetonitrile (CH$_3$CN) by OH. For CH$_3$CN we define a fraction (30%) to be converted to HCN, following Li et al. (2009) but alternatively we prescribe it as a purely chemical sink process in the troposphere, on top of the loss at the surface due to ocean uptake. This allows it to be used as a tracer for selected air-masses with the dominant emission sources being open burning fires (Li et al., 2000). Again, validation of these new tracers is not relevant to this study therefore not
presented here.

**Table 5**: Details related to the inclusion of HCN and CH$_3$CN in IFS(CBA), with rate expressions coming from Atkinson et al. (2004).

| Reaction | Rate expression | Comments |
|---|---|---|
| OH + HCN → | $1.2 \times 10^{-13} \times \exp(-400/T)$ | No products defined |
| OH + CH$_3$CN → 0.3HCN | $8.1 \times 10^{-13} \times \exp(-1080/T)$ | Products not completely defined |

**2.2 IFS(MOCAGE)**

The IFS(MOCAGE) chemical scheme, hereafter called IFS(MOC), is a merge of reactions of the tropospheric RACM scheme (Stockwell et al., 1997) with the reactions relevant to the stratospheric chemistry of REPROBUS (Lefèvre et al., 1994; Lefèvre et al., 1998). As in the IFS(CBA) implementation, IFS(MOC) uses a lumping approach for organic trace gas species. IFS(MOC) initial tropospheric RACM chemistry scheme was extended and now includes the sulfur cycle in the troposphere leading to the introduction of DMSO and NH$_3$ gas species
(Ménégoz et al., 2009; Guth et al., 2016). The current version of the IFS(MOC) chemistry scheme uses from now on 123 species, including long-lived and short-lived species, family groups and a polar stratospheric clouds (PSC) tracer, 319 gas-phase thermal reactions, 56 photolysis reactions and 14 heterogeneous reactions (9 for the stratosphere and 5 for the troposphere). The version adopted here is as that reported in Huijnen et al. (2019) with four major differences.

Firstly, in IFS(MOC), the formation of nitrate, ammonium and sulfur particles from NH$_3$, SO$_2$ and HNO$_3$ gaseous species is now activated in an analogous way to what is used in IFS(CBA). The nitrate and ammonium formation depends on resident sulphate concentrations (c.f. Huijnen et al., 2019; Rémy et al., 2019). This primarily affects the modeled NH$_3$ and HNO$_3$ atmospheric burdens, but indirectly also affects the other trace gases through
heterogeneous reactions.

Secondly, the formation of gaseous secondary organic aerosol precursors (SOG) from biogenic and anthropogenic and biomass burning volatile organic compounds (VOC) was implemented in IFS(MOC) following the simplified approach proposed by Spracklen et al. (2011). While biogenic sources (namely isoprene and monoterpene) are
provided by the IFS(MOC) chemistry scheme, anthropogenic and biomass burning emissions were scaled from



the corresponding CO emissions. In this simplified approach, only two Secondary Organic aerosol precursor Gas (SOG) low-volatility classes are considered, one for biogenic SOG and the other gathering both the anthropogenic and biomass burning contributions. As in IFS(CBA), this SOG chemistry is coupled in IFS(MOC) with the aerosol module by solving the equilibrium of the partitioning between gas and aerosol phase.

Recent developments in IFS(MOC) also include the modelling of the HCN and $CH_3CN$ tracers with chemical loss being limited to oxidation by OH and photolysis in the stratosphere, using the same rate data as those used in IFS(CBA) (see Table 5), but with no products defined. As already stated for IFS(CBA), validation of these new tracers is not relevant to this study and therefore not presented here.

Last, the heterogeneous scavenging on aerosol, cloud droplets and ice particles for $N_2O_5$, $HO_2$ and $NO_3$ has been fully updated, and made consistent with the IFS(CBA) configuration. For the sake of simplicity, the heterogeneous reaction probabilities ($\gamma$) are for most surfaces considered constant, as summarized in Table 6. The reaction probability $\gamma(HO_2)$ is computed following Thornton et al. (2008), taking the role of pH and partition between the gaseous $HO_2$ and the dissociated form ($O_2$) into account, adopting a constant pH of 5.5. It follows the description given in Huijnen et al. (2014), where further details are given.

**Table 6**: Heterogeneous $\gamma$ values used for prescribing conversion rates on atmospheric cloud droplets, ice and aerosols. 'T08' refers to Thornton et al. (2008).

| Particle type | $\gamma$ ($N_2O_5$) | $\gamma$ ($HO_2$) | $\gamma$ ($NO_3$) |
|---|---|---|---|
| Cloud droplets | 2.7*10-5 exp(1800/T) | T08 | NC |
| Ice | 0.02 | 0.025 | NC |
| Desert Dust | 0.01 | 0.06 | 0.01 |
| Sea-Salt | 0.02 | T08 | 0.01 |
| Organic Matter | 0.02 | T08 | 0.01 |
| Black Carbon | 0.01 | T08 | 0.01 |
| $SO_4$ | 0.02 | T08 | 0.01 |

## 2.3 IFS(MOZART)

The IFS(MOZART), hereafter referred to as IFS(MOZ), is based on the Model of Ozone and Related chemical Tracers (MOZART) mechanism (Kinnison et al., 2007; Emmons et al., 2010) and includes additional species and reactions from the Community Atmosphere Model with interactive chemistry, referred to as CAM4-chem, a component of the Community Earth System Model (CESM) (Tilmes et al., 2016).

The IFS(MOZ) version used here is comparable to that described and evaluated in Huijnen et al. (2019) with few additional updates. The TOLUENE lumped aromatic used in the previous version has been replaced by the separate species BENZENE, TOLUENE, and XYLENE, allowing accounting for their different lifetimes and oxidation products. The current version also includes formic acid (HCOOH) and ethyne ($C_2H_2$) and their oxidation products. The reaction rates have been updated to follow the recommendations from IUPAC (http://iupac.pole-ether.fr; last access: 21 Sept 2021) or JPL-2015 (Burkholder et al., 2015). Updates have also been introduced to the photolysis look-up table, which includes explicit calculations of photolysis frequencies for most of the IFS(MOZART) species.

The heterogeneous chemistry in the troposphere is implemented according to the corresponding module from IFS(CBA) to account for heterogeneous uptake of $N_2O_5$, $HO_2$ and $NO_3$ on aerosols as described in the previous section. However, the heterogeneous uptake on ice and cloud droplets is currently not included in IFS(MOZ).

Finally, nitrate, ammonium and sulfur particle formation have been introduced in IFS(MOZ) analogous to IFS(CBA) and IFS(MOC) (Rémy et al., 2019), involving gaseous $SO_2$, $NH_3$ and $HNO_3$, as described above.



## 2.4 Setup of model simulations

The model simulations using the new developments described above have been performed with IFS cycle 47r1.
The simulations have been run using 137 vertical levels and a horizontal resolution of $T_L255$, corresponding to ~0.7° x 0.7°, excluding data assimilation of atmospheric composition.

The simulations evaluated here are executed as a series of 24 h hindcasts, daily initialized at 0h UTC from ERA5 meteorology (Hersbach et al., 2020). A 30-minute time step was used. A four-month spin-up period is employed to allow the system to reach chemical equilibrium during September to December 2013, after which an 18-month
simulation was performed to allow the use of winter-time measurements for deriving some seasonality in the analysis. To limit the volume of data, outputs of 3D chemical fields are archived at a 3-hourly time interval and subsequently interpolated to the relevant time stamp of each measurement. The experiments are archived with experiment identifiers b28w for IFS(CBA), b0ov for IFS(MOC) and b0yw for IFS(MOZ).

For comparison, the CAMS Reanalysis (CAMSRA), which includes data assimilation of $O_3$, CO, $NO_2$ and
Aerosol Optical Depth (AOD), uses IFS cycle 42r1. CAMSRA was run at the same horizontal resolution ($T_L255$) as the experiments presented here, but with only 60 model levels, a model top of 0.1hPa (Inness et al., 2019). Different emission estimates were also employed for the CAMSRA dataset meaning that interpretation of differences need to be done with care.

The emissions adopted in this configuration are taken from CAMS_GLOB_ANT v4.2-R1.1 (Granier et al, 2019;
https://eccad.aeris-data.fr/), which is a modification of the v4.2 global anthropogenic emission dataset used operationally in CAMS, with reduced emission strength over China. Biogenic emissions are taken from the CAMS_GLOB_BIO v1.1 dataset (Doubalova et al., 2018; http://eccad.aeris-data.fr/), and biomass burning emissions are taken from GFAS v1.2 (Kaiser et al., 2012) which are applied using the methodology as described in Ye et al. (2021). As GFAS does not provide separate estimates for HCN and $CH_3CN$ emissions, we scale
them from the CO emissions. Other (oceanic, natural) emissions are taken from the standard configuration in IFS (see Huijnen et al., 2019).

Apart from Biomass Burning (BB) and $SO_2$, all emissions are applied in the lowest model level. Information on emission totals used in the simulations is given in Table 7 for IFS(CBA). This corresponds essentially to the emissions as adopted in IFS(MOZ) and IFS(MOC), with small variations in the partitioning of some of the
lumped VOC's such as the higher alkanes, alkenes and aldehydes, as well as the aromatics. When comparing emission totals with those used in Huijnen et al. (2019) main differences are 10% lower primary emissions for anthropogenic CO and $SO_2$, an 20% increase in anthropogenic $NH_3$, approximately equal $NO_x$ emissions and most importantly, a 20% reduction in $C_5H_8$. Some of these changes with respect to the anthropogenic contributions are furthermore due to the choice of another evaluation year and trends in the annual emission
estimates.

**Table 7**: Global annual mean emissions for 2014 as adopted in current IFS(CBA) model simulations [Tg species/year]. For NO, also the aircraft and lightning $NO_x$ emissions are given as anthropogenic and biogenic contributions, respectively

| Species | Anthropogenic | Biogenic/Oceanic | Biomass Burning |
|---|---|---|---|
| CO | 532 | 65/20 | 355 |
| NO | 71.9 + 2.1AC | 10.6 + 10.3 LiNO | 9.4 |
| HCHO | 4.4 | 3.3 | 5.4 |
| $CH_3OH$ | 14 | 99 | 11 |
| $C_2H_6$ | 4.8 | 0.3/1.0 | 2.5 |
| $CH_3CH_2OH$ | 2.3 | 13.6 | 0. |
| $C_2H_4$ | 6.9 | 21.9/1.4 | 4.8 |





| | | | |
|---|---|---|---|
| $C_3H_8$ | 6.1 | 0.03/1.3 | 0.3 |
| $C_3H_6$ | 3.2 | 13/1.5 | 3.3 |
| $CH_3CHO$ and higher aldehydes | 9.2 | 15.9 | 5.0 |
| $CH_3COCH_3$ | 6.7 | 32.5/27.8 | 3.0 |
| butanes and higher alkanes | 40.0 | 0.8 | 0.3 |
| butenes and higher alkenes | 2.5 | 0.5 | 0.8 |
| toluene | 17.1 | 1.1 | 7.9 |
| xylenes | 11.1 | 0 | 8.1 |
| $C_5H_8$ | 0 | 381 | 0 |
| terpenes | 0 | 78 | 0 |
| $SO_2$ | 89.7 | 14.8 | 1 |
| DMS | 0 | 38.0 | 0 |
| $NH_3$ | 53.8 | 2.3/8.1 | 10.8 |
| HCN | 1.18 | 0 | 1.7 |
| $CH_3CN$ | 0.44 | 0 | 1.14 |

## 3 Observations


In this section we provide an overview of all the observational data used to evaluate the performance of the three IFS versions for the years 2014 and 2015. Figure 1 shows the location of all the measurement stations and regions covered by the aircraft campaigns utilized for assessing the different versions of the IFS.

### 3.1 Surface flasks and soundings for CO and $O_3$

For tropospheric $O_3$ we use seasonal composites from four individual sites from the World Ozone and Ultraviolet Radiation Data Center (WOUDC; http://woudc.org), namely Yarmouth (43.9°N, 66.0°W), Boulder (40.0°N, 105.0°W), Trinidad Head (41.1°N, 124.2°W) and Huntsville (34.7°N, 86.7°W). Each station samples a different

region, allowing the assessment of performance at continental scale, with an associated error of around ±10% (Komhyr et al., 1995; Steinbrecht et al., 1998).

For evaluating the surface mixing ratios of $O_3$, we compare monthly mean mixing ratios against corresponding values taken from three measurement sites at Trinidad Head (41.1°N, 124.2°W), Boulder (40.0°N, 105.0°W) and Niwot Ridge (40.1°N, 106.5°W). These observations are taken from the long-term measurement network

maintained by the NOAA Earth System Laboratories (ESRL) Global Monitoring Laboratory. Correspondingly, for CO we use surface observations from the ESRL stations Park Falls (LEF, 45.9°N, 90.4°W), Niwot Ridge (NWR, 40.1°N, 105.6°W) and Key West (KEY, 25.6°N, 80.2°W). These observations have an associated uncertainty of between 1-3 ppb (Novelli et al., 2003).

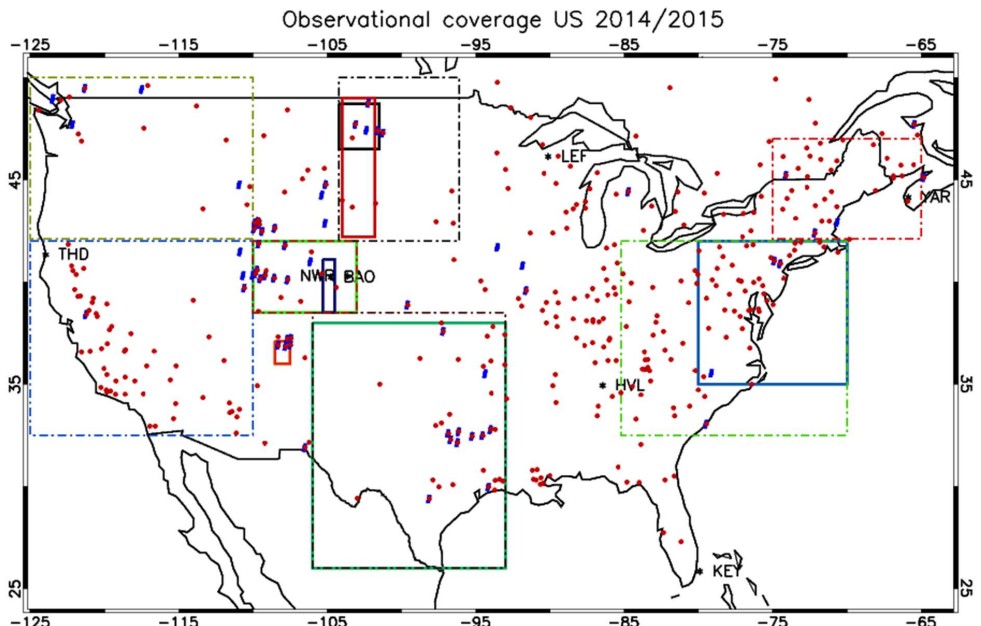

**Figure 1**: The location of surface measurement stations and areas sampled during the numerous aircraft campaigns. Key to air-shed: ― ND (May 2014), ― NM (June 2014), ― Col (July/Aug 2014), ― Col (July/Aug 2014; Mar/Apr 2015), ― VN (Feb/March 2015) and ― Texas (Apr 2015). The areas outlined by the dashed-dot borders are associated with the regions for which the AirNow observations are analysed, with the reader being referred to Table 10. The individual stations for the AirNow network are shown as the red dots ($O_3$) and blue dots ($NO_2$) for rural stations only, where for clarity we omit the suburban/urban stations.

### 3.2 Aircraft Observations

Figure 1 also shows the predefined air-sheds used to evaluate the lower troposphere (LT) over various regions across the US during 2014 and 2015. There is a paucity of aircraft data over the western part of the US, as most aircraft campaigns focus on industrial regions or those influenced by energy production techniques such as fracking for the chosen evaluation year. Data from the following campaigns are used:

- TOPDOWN (May/June 2014): Springtime campaign focusing on emissions from rural fracking sites located in North Dakota and New Mexico: (https://csl.noaa.gov/groups/csl7/measurements/2014topdown/ ; last access: 28 Sept 2021).
- Front Range Air Pollution and Photochemistry Éxperiment (FRAPPÉ) / Deriving Information on Surface conditions from Column and Vertically Resolved Observations Relevant to Air Quality (DISCOVER-AQ) (July/Aug 2014): Summer campaign investigating $O_3$ exceedances due to regional VOC and $NO_x$ emissions in Colorado (Dingle et al., 2016; McDuffie et al., 2016; Flocke et al., 2019). More details related to DISCOVER-AQ can be found at https://www.nasa.gov/larc/2014-discoveraq-campaign/ (last access: 28 Sept. 2021).
- Wintertime INvestigation of Transport, Emissions, and Reactivity (WINTER): Winter campaign over the East Coast to investigate $O_3$ and $NO_x$ chemistry (Feb/Mar 2015) over Virginia and North Carolina (McDuffie et al., 2018; Jaeglé et al., 2018).
- Studying the Atmospheric Effects of Changing Energy Use in the U.S. at the Nexus of Air Quality and Climate Change (SONGNEX): Spring study on effects of shale and natural gas extraction on



air quality across the US (https://csl.noaa.gov/projects/songnex/; last access: 28 Sept 2021, Peischl et al., 2018; Koss et al., 2017).

**Table 8:** Details of the various aircraft measurement campaigns in 2014/2015 used in this study.

| Campaign | Date | Trace species | US state | Regional domain (Lat, Lon) |
|---|---|---|---|---|
| TOPDOWN | May/June 2014 | $O_3$, CO | North Dakota (ND) North Mexico (NM) | 46.5-48.7°N, 101.5-104.2°W 36-37.1°N, 107.5-108.5°W |
| DISCOVER-AQ | July/August 2014 | $O_3$, CO, $NO_2$, NO $HNO_3$, PAN | Colorado (Col) | 39.5-40.6°N, 104.5-105.3°W |
| FRAPPE | July/August 2014 | $O_3$, CO, $NO_2$, NO $HNO_3$, PAN | Colorado (Col) | 38.5-42°N, 103-110°W |
| WINTER | Feb/March 2015 | $O_3$, CO, $NO_2$, NO $N_2O_5$, $HNO_3$, PAN | North Carolina (NC) Philadelphia (PL) Washington(WS) Virginia (Vir) | 38.5-42°N, 70-80°W |
| SONGNEX | March/April 2015 | $O_3$, CO, $NO_2$, NO $HNO_3$, PAN | North Dakota (ND) Colorado (Col) Texas | 42-50°N, 101.8-104°W 38.5-42°N, 103-110°W 26-38.5°N, 93-106°W |

370

The aircraft campaigns, whose regional coverage and trace species are given in Table 8, are either segregated into different legs for different US states or sample air over a wide area for different days. In the latter case we segregate spatially during the analysis to provide more regional coverage. Lower limits for the trace gases $NO_2$ and $NO_y$ are placed on the observations as determined by the detection limits of the instrumentation of 40 ppt as stipulated in the data files. For $NO_2$, $HNO_3$ and PAN dual measurements are available from different instruments, which are merged to increase the available sampling frequency. For $N_2O_5$, a maximum of 1.3 ppb is placed on the observations for the comparisons presented in to avoid spurious values.

We interpolate 3-hourly data output from the model simulations using the time, geolocation and pressure of the observations, then average over predefined pressure bins ensuring that there are a sufficient number of observations per bin. We also calculate mean bias statistics for the LT using selected pressure tops of 815 hPa (Col), 900 hPa (EC), 880 hPa (ND) and 965 hPa (Texas) accounting for the variable orography and to include a sufficient number of points as to be statistically robust. The resulting values are presented in the Supplementary Material as Tables S1 through to S4 ($O_3$, CO, $NO_2$, NO).

### 3.3 Surface air quality networks for $O_3$, CO and $NO_2$

For evaluating surface concentrations of the model output we exploit observations of $O_3$, CO and $NO_2$ from the AirNow (www.airnow.gov; last access: 28 Sept 2021) air quality observations network. Within the AirNow centralized system, hourly measurements of $O_3$, $NO_2$, CO, PM2.5 and PM10 are collected from measurement locations across the U.S., submitted by state or local monitoring agencies and made available in near real time, after preliminary data quality assessments have been performed. Here we use data collected over the 2014 period for the designated domains, which includes 570, 222 and 123 monitoring stations for $O_3$, $NO_2$ and CO, respectively. The list and extension of the sub-domains over which statistical scores are provided is given in Table 9, together with the number of stations used in the current study. Each station is designated as classification being either urban, suburban or rural depending on the location of each station in order to differentiate between clean and polluted sampling locations. Only clean background comparisons are shown due to the difficulty of global models representing small scale gradients in concentrations.





Additionally, a filtering procedure was applied to the data before computing the daily mean values to remove stations where the time series displayed more than 50% of identical values, denoting a failure in the measurement sensor. Moreover, for CO the observational values < 50 μg m$^{-3}$ and >1200 μg m$^{-3}$ were filtered out at the rural stations to avoid spurious instrumental effects. Model 3-hourly outputs were spatially interpolated from the model grid to the stations network. Time-series of daily mean composites were then obtained by computing the daily mean values across stations with identical station classification for each domain, using both the observed and simulated data during 2014, providing a spatial mean of the concentrations. Details on the accuracy of the instrumentation used across the network can be found in Williams et al. (2014).

**Table 9**: Details of the various regions defined for assessing simulated surface concentrations for $O_3$, CO and $NO_2$ using data from the AirNow measurement network. The total number of individual stations for each respective species is given in the right most column.

| Region | Regional domain (Lat/Lon) | Total no. of stations ($O_3$/CO/$NO_2$) |
|---|---|---|
| Maine | 42-47°N, 65-75°W | 55/10/37 |
| North/South Dakota | 42-50°N, 96.1-104.2°W | 18/2/7 |
| Colorado | 38.5-42°N, 103-110°W | 27/9/12 |
| Washington/Oregon | 42-50°N, 110-125°W | 57/23/46 |
| California/Nevada | 32.5-42°N, 110-125°W | 221/47/64 |
| Texas | 26-38.5°N, 96.1-104.2°W | 47/4/20 |
| East Coast | 32.4-42°N, 70-85.2°W | 145/28/36 |

**3.4 Surface network for HNO$_3$**

Finally in order to evaluate the near-surface concentrations of $HNO_3$, we use the near-surface mixing ratios taken from the Clean Air Status and Trends Network (CASTNET; https://www.epa.gov/castnet ; last access 30 Sept 2021), which provides filter pack measurements at weekly intervals for 87 individual sites located in rural locations away from strong $NO_x$ emission sources. This dataset is therefore a measure of the chemical processing which occurs in transported air-masses at regional scale. The inferred mixing ratios result from the application of the Multilayer Model (Finkelstein et al., 2000) to these samples, which uses site specific meteorological parameters to determine deposition values. We only use the near-surface mixing ratios here, as assessing the wet deposition component of the chemical modules is beyond the scope of this manuscript and more associated with the aerosol modules applied in IFS. For brevity, we aggregate and compare seasonal mean values of the mixing ratios as interpolated from the model data using the location and height of each station for correct interpolation. It should be noted that from an intercomparison of gaseous $HNO_3$ values, CASTNET data were found to be typically lower than those of other measurement networks (Lavery et al., 2009). However, CASTNET data composites are typically used for assessing the performance of deposition processes in global transport models and is an established dataset.

## 4 Results

### 4.1 Tropospheric O$_3$


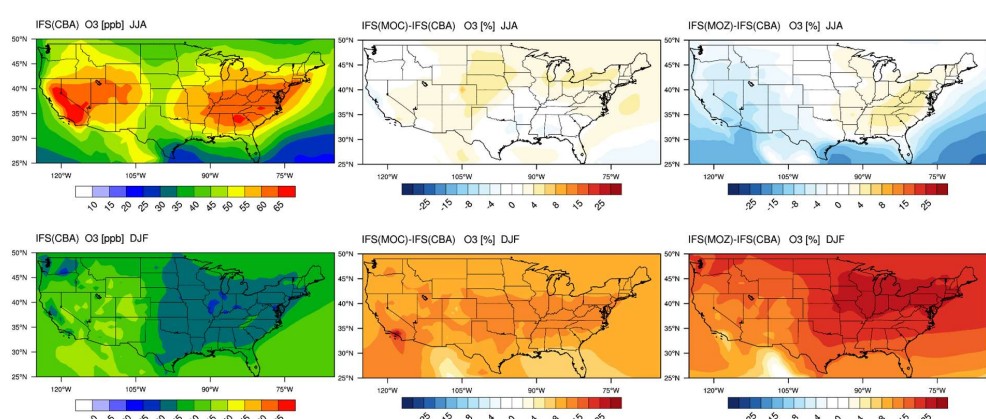

**Figure 2:** The horizontal seasonal mean for tropospheric O$_3$ below 1km over the US domain for JJA 2014 (top) and DJF 2014/2015 (bottom).

The seasonal horizontal mean distribution of tropospheric O$_3$ in IFS(CBA), and the associated relative differences for the IFS(MOC) and IFS(MOZ) against IFS(CBA), are shown for seasons December-January-February (DJF) and June-July-August (JJA) over the US in Figure 2. We select all levels below 1km for calculating the seasonal means, which allows a direct comparison for coastlines and elevated regions (namely Colorado). For DJF we choose 2014-2015 data to be fully conversant with the validation of the vertical profiles (see later) and when the
system has achieved chemical equilibrium for longer lived species.

For JJA, the seasonal distribution shows higher mixing ratios exceeding 65 ppb at both the East and West coasts driven by regional NO$_X$ emissions. There is a minimum in Central rural regions centred around Colorado, with a variability of between 35-45 ppb in IFS(CBA). Examining regional differences shows that when enhancements in (near-)surface O$_3$ typically occur, both IFS(MOC) and IFS(MOZ) show a ±5% variability with respect to
IFS(CBA), with maximal differences on the east coast of +4-6%. Over the surrounding oceans, IFS(MOZ) has a decrease in mixing ratios of between 5-10 ppb due to reduced transport. During DJF the model shows a weak west-east gradient with a lower range of mixing ratios of 30-50 ppb. For DJF, the range of the differences is substantially higher (~8-35%) than for JJA, with IFS(MOZ) having a significant excess in O$_3$ towards the east coast with respect to IFS(CBA). Therefore, under identical NO$_X$ emissions, the O$_3$ production efficiency via the reactive NO$_x$ cycle
is highest for IFS(MOZ) indicating a lower rate of termination towards NO$_y$ (See Sect. 6). This subsequently increases mixing ratios of the hydroxyl radical (OH) from the primary production term involving photolysis of O$_3$ in the presence of water vapour (H$_2$O) (see Fig S1).

Evaluations against seasonal O$_3$ sonde composites of all available measurements for JJA and DJF located across the US for the lower troposphere, as shown in Figure 3, also exhibits a signature of the seasonal mean differences
as discussed for Fig. 2. For JJA there are positive biases for stations on the east coast, whereas for Trinidad Head (at the west coast) the agreement is more favorable. The highest bias is shown for Huntsville in Texas of between 10-20 ppb, where the station is affected by the transported plumes from nearby cities (Newchurch et al., 2003). In general, the correct profile shape is captured for most sites, except for Trinidad Head where a steep gradient is observed. For DJF, there is more variability across the chemical modules where IFS(MOZ) has the highest
mixing ratios towards the east coast leading to a positive bias of 15 ppb at Yarmouth. Again, profile shapes are captured well across the stations.



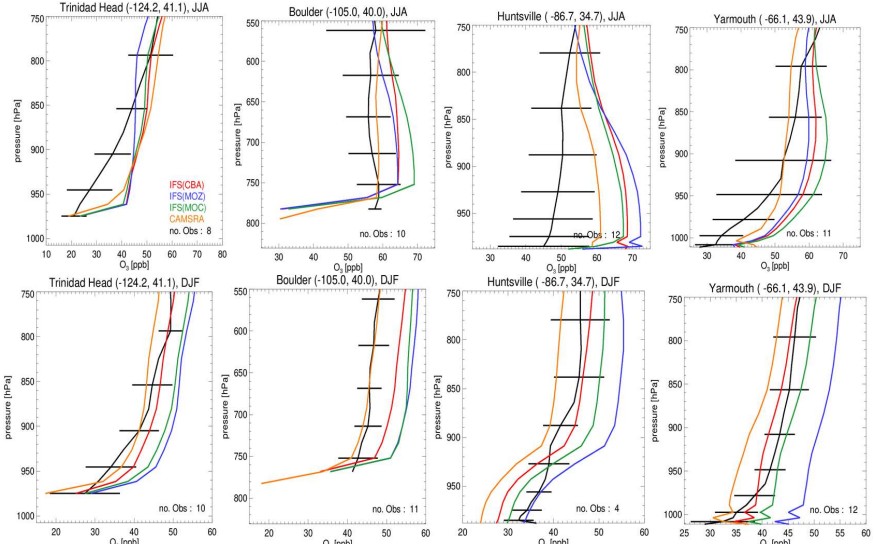

**Figure 3**: Seasonal O$_3$ sonde comparisons for seasons JJA (top) and DJF (bottom) for the lower troposphere. Stations shown are Trinidad Head, Boulder, Huntsville and Yarmouth which are situated across the US. The observational composites are shown in black, with the colour key provided in the top left panel. The geolocation of each site is given in the title of each panel and number of observations used bottom right.

In Figure 4 the corresponding annual cycle in monthly mean mixing ratios for surface O$_3$ at the three continuous
measurement sites available in the US are shown, with two (BAO and THD) being co-located in vicinity of the sonde measurements. For THD, which is situated at the outermost west coast of the US, the observed monthly mean mixing ratios are very low. This indicates the absence of significant local NO$_x$ emissions, therefore being influenced by land-sea air movements allowing the sampling of clean pacific air (Oltmans et al., 2008). For most months in THD and NWR, there is a bias of up to 100% across the different chemical modules with a more muted
amplitude in monthly variability than observed, indicating difficulties of the global model configuration towards capturing the correct seasonal cycle. Apart from model resolution effects, the transport of air from out of the US could be too efficient as described by the transport processes, or the production and loss over the ocean surface is not in the correct equilibrium. Comparing IFS(CBA) and CAMSRA there has been an increase of between 2-10 ppb in surface O$_3$ mixing ratios, increasing the simulated biases somewhat.

Even though the stations BAO and NWR are relatively close together there is a diverse shape in the annual cycle showing the influence of regional NO$_x$ chemistry, meteorology and station height (1.5km versus 2.9km above sea level). Transport of pollutants from the Denver region affects O$_3$ mixing ratios at NWR (Chin et al., 1994; McDuffie et al., 2016), thus representative of a chemically aged air-mass. The BAO site is also influenced by anthropogenic emissions with measurements sampling the boundary layer (Gilman et al., 2013; McDuffie et al.,
2016). At BAO the annual cycle and maximal mixing ratios are captured well, with IFS(CBA) performing better than CAMSRA during the winter months. For NWR, which samples air above the surface, there is no strong annual cycle in the measurements, whereas for the CAMSRA and the mini-ensemble a maximum occurs during JJA with a bias of between ~50-70%.

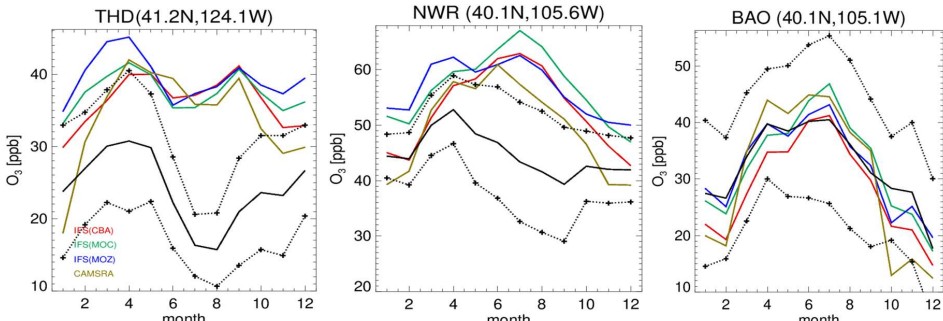

**Figure 4**: Comparisons of surface $O_3$ measurements for 2014 for the three GFDL stations situated in the US, namely: Trinidad Head (THD), Niwot Ridge (NWR) and Boulder (BAO). The color key to the various ensemble members is given in the left panel. The observational means and the corresponding 1-σ variability are shown as the black solid and dashed lines, respectively.

The variability in the daily mean values in surface $O_3$ simulated by the chemistry modules are evaluated against
regional composites of measurements taken from the AirNow network assembled for 2014, see Figure 5. This is done for the regions defined in Table 10 (Sect. 3), where only those stations which are classified as rural (i.e. away from urban conurbations) are used. Model performance is regionally dependent, with larger biases for regions exhibiting strong seasonality on the meteorology and the length of day (e.g. California-Nevada). During the first few months of 2014 a significant mean positive bias exists for more northerly regions across the chemical
modules, with IFS(MOZ) (IFS(CBA)) typically being 40-60 μg m$^{-3}$ (0-20 μg m$^{-3}$). For CAMSRA, the corresponding mean biases are negative upto 40 μg m$^{-3}$ e.g. Washington-Oregon. For Colorado and California-Nevada a similar seasonal behaviour occurs, with CAMSRA exhibiting higher negative mean daily wintertime biases, whereas for e.g. the East Coast and Texas limited seasonality is seen with a smaller divergence across the chemical modules and CAMSRA. For more southerly regions the mean daily biases across chemical modules
are smaller, with those for CAMSRA turning positive (up to 20 μg m$^{-3}$). The associated correlation between observations and simulations ranges between r=0.4-0.8 for Maine and the East Coast across stations, whilst being anti-correlated in California-Nevada (r ranging from 0 to -0.4), with strong similarities between IFS(CBA) and CAMSRA. For boreal summertime, significant positive biases occur for all chemical modules at both coasts (40-80 μg m$^{-3}$), with Colorado exhibiting smaller negative biases (upto 20 μg m$^{-3}$) due to strong regional NO and
VOC mixing ratios (more daytime titration and radical chemistry). In terms of correlation, there is a weak-to-moderate positive correlation for Maine and Texas (r=0-0.6), whereas for other regions there are negative correlations ranging from R ranging from -0.2 to -0.4 showing the complex nature in the relationship between $NO_x$ precursors and the efficacy in forming tropospheric $O_3$. For urban stations the observations show lower concentrations during boreal wintertime (20-30 μg m$^{-3}$) than boreal summertime (50-60 μg m$^{-3}$), with significant
daily variability. For these polluted stations generally there is a high positive bias (70-120%) across regions during the summer months indicating too efficient $O_3$ production and/or too little transport and mixing, as seen in previous studies (Huijnen et al, 2019). The description of small-scale urban chemical processing with influence from street canyons and local point sources (factories, processing) is not included in the current CAMS description thus leading to such biases. However, the seasonal variability in the outflow from such polluted
regions is described well enough not to lead to corresponding biases for the rural comparisons.



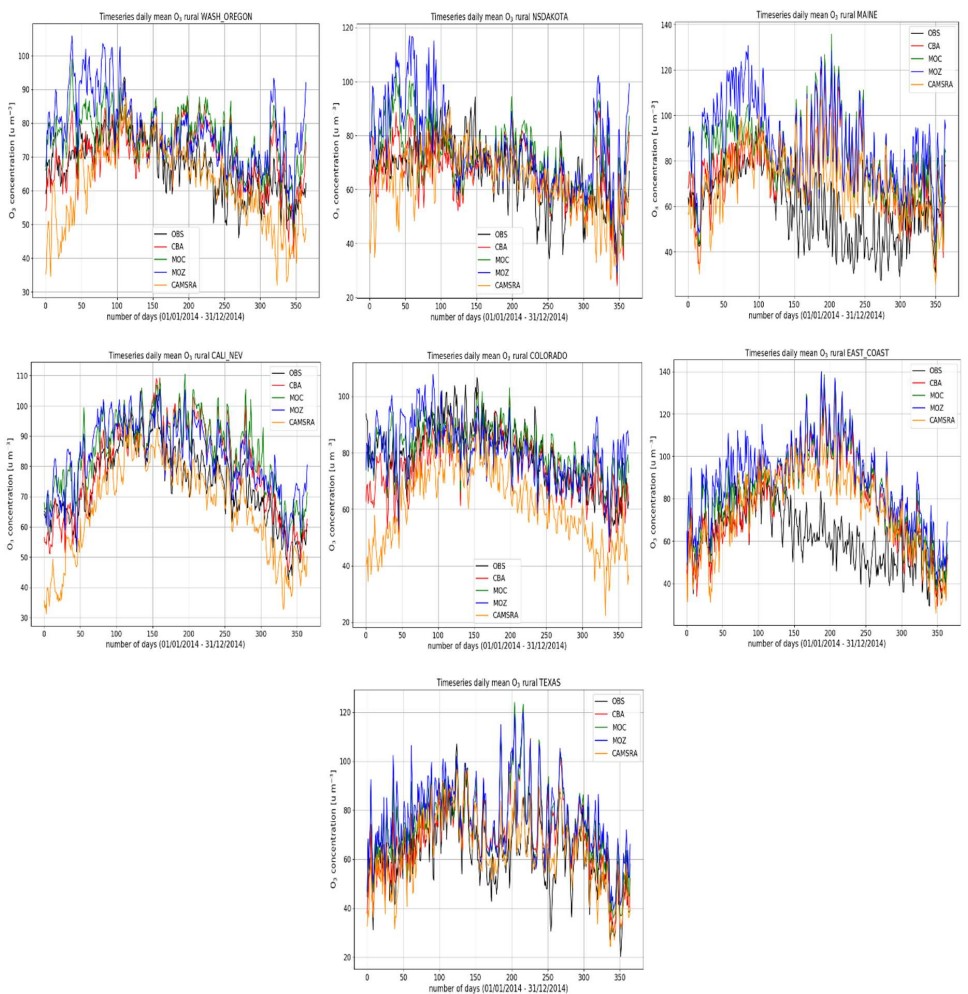

**Figure 5:** Comparisons of the daily mean variability surface O₃ for 2014 against regional composites assembled from measurement sites for rural locations contributing to the AirNow network. Regions shown (top left to bottom right) are Washington-Oregon, North/South Dakota, Maine, California-Nevada, Colorado, Eastern Coast, and Texas.

Vertical profiles in the LT across the various US regions are shown in Figure 6, with the boundaries of various regions given in Fig. 1 and Table 3. The corresponding mean bias statistics are documented in Table S1. The shallow to moderate gradients in LT O₃ seen in the observations are simulated relatively well across the various chemistry modules, albeit with a changing variability and bias across the regions. For summertime, most comparisons are for the Colorado region over Denver and the surrounding area, where O₃ production is heavily influenced by oil and natural gas production and Industry (Cheadle et al., 2017) exhibiting mean values 60-65 ppb for the (near-)surface. During July the region experienced a strong cyclonic front, whereas in August non-cyclonic conditions occurred (Vu et al., 2016) resulting in different transport dynamics for each period although little impact is seen in the mean value. Here, the monthly negative biases are of the order of -2 to -10 ppb, indicating too low regional $NO_x$ emissions (c.f. Table S3) and showing that the persistent positive bias seen at more globally remote regions (Huijnen et al., 2019) does not occur during summertime for more polluted urban regions. For CAMSRA more significant negative biases exist of between 10-15 ppb.



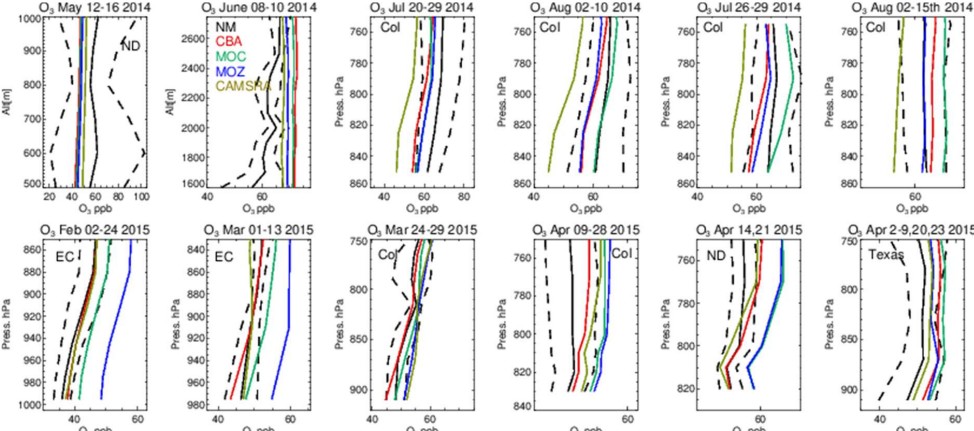

**Figure 6**: Comparisons of lower tropospheric $O_3$ profiles for 2014/2015 against composites of aircraft measurements for the regional domains shown in Figure 1. The 1-σ standard deviation of the mean for the observational values are shown as the dashed line. Campaigns shown (top left to bottom right) are TOPDOWN, DISCOVER-AQ, FRAPPE, WINTER and SONGNEX.

The evaluation during boreal wintertime over the East Coast, which has lower observed $O_3$ mixing ratios of 38-45 ppb, reveals there is more divergence across the chemistry modules under these conditions, with IFS(MOZ) showing high positive biases of ~10 ppb, whereas IFS(CBA) captures the observational mean profile well within a few ppb. This leads to high oxidative capacity for IFS(MOZ) which has a subsequent impact on tropospheric CO (see next section). This evaluation highlights differences in model performance in the lower troposphere compared to those presented for the surface $O_3$ analysis. In that vertical mixing has occurred means that this analysis is not subject to the representation of locational issues with measurement sites (local emission sources such as roads, the effects of building on transport, etc.). For springtime over Colorado mixing ratios are somewhat lower than summertime (~50 ppb), where IFS(CBA) shows a small negative biases of a few ppb, with the positive bias for IFS(MOZ) persisting (5 ppb). For Texas, a positive bias of 5 ppb occurs across chemistry modules, with the CAMSRA dataset having similar biases throughout regions during springtime.

### 4.2 Tropospheric CO

The corresponding US continental distribution for seasonal mean tropospheric CO for IFS(CBA) for JJA and DJF are shown in the top and bottom left panels of Figure 7, respectively. A distinct east-west gradient exists with ~50% higher mixing ratios towards the East coast reaching ~150-160 ppb. No distinct burning regions are visible towards the west coast associated with comparatively low BB activity for 2014 in the US (Petetin et al., 2018). The two other chemistry modules have consistently lower mixing ratios under identical primary CO emissions, indicating differences in the chemical production rate from the oxidation of formaldehyde and higher Volatile Organic Compounds (VOC), combined with differences in the chemical lifetimes as a result of OH variability (Huijnen et al., 2019). For IFS(MOZ) we diagnose a comparatively low tropospheric CO burden associated with a fast oxidation rate due to higher mixing ratios of OH in the LT of between 20-50% (c.f. Fig S1). This is directly associated with the higher $O_3$ (c.f. Fig 2) in IFS(MOZ). For DJF, a much shallower continental gradient exists with average mean mixing ratios of between 100-120 ppb, with IFS(MOC) and IFS(MOZ) again exhibiting negative differences compared to IFS(CBA). Signatures of increased pollution are visible over large urban centers in e.g. California and Washington state, in line with figures presented for LT $O_3$ related to regional $NO_x$ emissions (See next section).


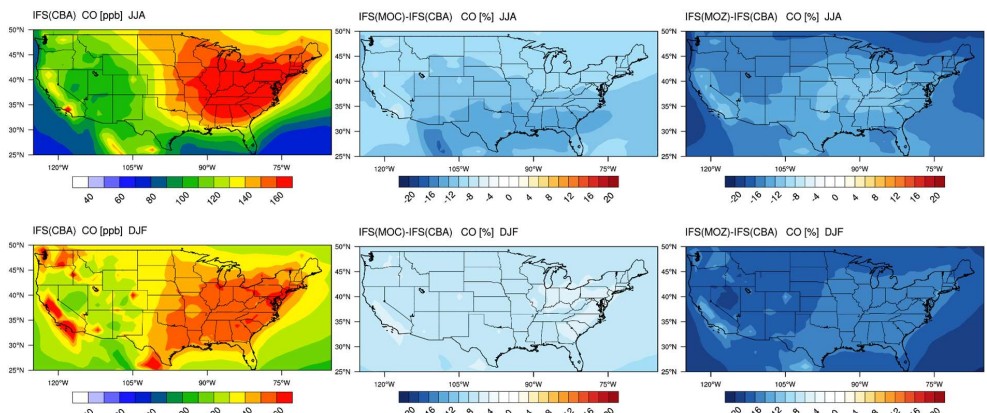

**Figure 7**: The horizontal mean cross-section of CO below 1km over the US domain for JJA 2014 (top) and DJF (2014/2015).

The seasonal cycle in surface CO mixing ratios is compared against monthly mean composites from the ESRL observational network. The seasonal cycle is somewhat determined by the regional CO emissions as exemplified
by the increase observed for July at Park Falls (LEF), which is possibly due to local BB events for this year. A signature exists at NWR from the chemical processing of polluted air masses from the Denver region during summertime, where all members show similar positive biases of ~ 25%. For Key West (KEY) the seasonal cycle is representative of the outflow from the continental US, where all members capture the monthly variability quite well. For many of the individual months, IFS(CBA) shows a mild improvement when compared to the CAMSRA
dataset, which shows a persistent positive bias especially during boreal wintertime. IFS(MOZ) exhibits the largest negative bias across stations in line with the lower mixing ratios shown in Fig.7. In some instances, biases are of the order of 100% especially for winter months where OH is typically lower (c.f. Fig S1), and pollution is less mixed into the free troposphere.

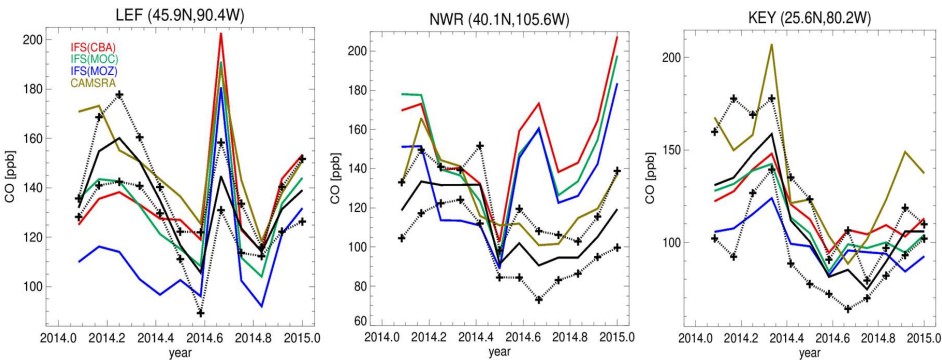

**Figure 8**. Comparisons of monthly mean surface CO mixing ratios at three separate surface measurement sites across the US. The observational monthly means with the variability are shown in black.





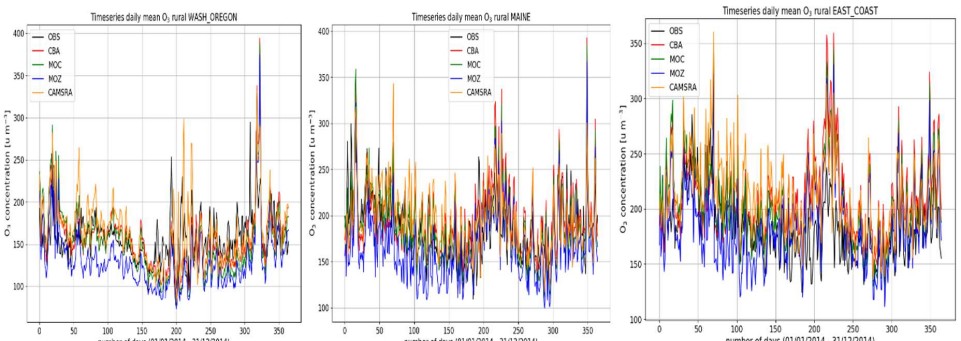

**Figure 9**. Comparisons of the daily variability of surface CO for 2014 against regional composites assembled from measurement sites participating in the AirNow network. Regions shown (left to right) are Washington-Oregon, Maine and the Eastern Coast as determined by the number of available measurements for the rural classification.

More extended regional composites for surface CO have been assembled using data available from the AirNow measurement network, with three regional comparisons of the daily variability in surface CO at rural locations being shown in Figure 9. The number of stations used for the comparison is lower than that used for the other species as limited by data availability (c.f. Table 10). No significant annual cycle is present in any of the domains shown. For boreal wintertime, again IFS(MOZ) (IFS(CBA)) exhibits the largest (smallest) negative bias as seen for the ESRL comparisons (Fig. 8 above), with a daily mean bias of between 80-125 µg m⁻³ (50-80 µg m⁻³) and only a moderate correlation across chemical modules (R = 0.2-0.4). CAMSRA exhibits the lowest biases of around 25-50 µgm⁻³, likely aided by assimilation, with an improved correlation (R = 0.3-0.6). For boreal summertime, biases between IFS(MOZ), IFS(CBA) and CAMSRA are similar and around 20-40 µgm⁻³, again agreeing with the ESRL comparisons for this season. The seasonal correlation in CO concentrations is similar to that for boreal wintertime, with Pearson's R < 0.6 for any of the stations used in the evaluations. For the urban environment higher CO concentrations are observed (typically 200-300 µg m⁻³; not shown) resulting in significant negative biases across the various chemical modules, where sub-grid maxima from strong point sources is diluted within any typical grid cell. For polluted scenarios, CAMSRA does not exhibit any significant difference from the other simulations.

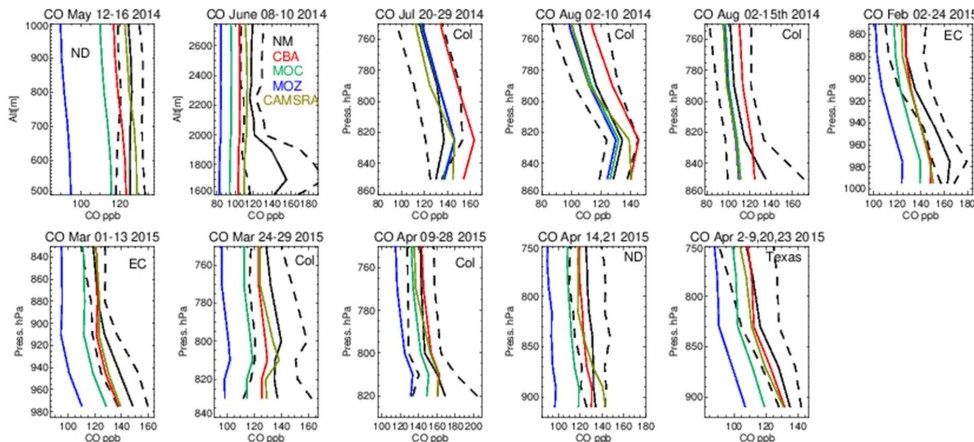

**Figure 10**. As for Fig. 6 except for tropospheric CO. Campaigns shown (top left to bottom right) are TOPDOWN, DISCOVER-AQ, FRAPPE, WINTER and SONGNEX.

Tropospheric CO profiles in the LT are compared against the corresponding aircraft composites across the various campaigns in Figure 10. As for O₃, the shape of the vertical profiles is captured well with a significant variability in mixing ratios across the ensemble as shown in the seasonal mean comparisons in Figure 7. For NM the convex



bulge seen in the LT is not captured by the ensemble most probably due to missing emissions from energy production (fracking) which was the focus of this measurement campaign. High negative biases of 35-60 ppb occur for June 2014, showing a significant underestimation in either primary emissions or VOC precursors (no measurements available) for this region. For boreal summertime over Colorado mean values between 100-130 ppb exist, where IFS(CBA) exhibits positive biases of 5-20 ppb (c.f. Table S2). Comparing IFS(CBA) against CAMSRA shows there is only a modest difference in CO, with slight increases in the associated negative biases. Biases presented in Table S2 shows that, assuming comparable data quality, the influence of local effects related to positioning and selection of stations can result in more extreme biases. For the aircraft composites surface effects are not so important where mixing in the boundary layer provides a more homogenized value. For wintertime and springtime the higher observed mean values of 130-175 ppb are not captured by the mini-ensemble with underestimates of 10-40 ppb depending on the month and region suggesting too low primary emissions over a wide area, with CAMSRA exhibiting the lowest biases.

One dominant precursor for the chemical production of CO is the oxidation of formaldehyde ($CH_2O$) e.g. Zeng et al. (2015). The corresponding comparisons of $CH_2O$ for all chemical modules are shown in the Supplementary Material in Figure S4, with associated biases in Table S3. In the LT during boreal summertime conditions, $C_5H_8$ acts as a dominant source of $CH_2O$ resulting in high mixing ratios over Colorado of 1.7-2.5 ppb. Generally, all chemical modules exhibit negative biases, except for the CAMSRA dataset, associated with the higher $C_5H_8$ emissions which are applied (MEGAN-MACC; (Sindelarova et al., 2014)). There is a higher variability associated with the simulations than that observed showing the sensitivity towards both photolysis and dissolution into cloud droplets, which introduces complications in short-term (daily) modelling of $CH_2O$. Note that the higher $CH_2O$ in CAMSRA is not directly correlated with higher CO, due to effective CO data assimilation being applied.

During wintertime biogenic fluxes are low due to the seasonality in biogenic activity, thus CO comparisons shown for February and March can be considered to be representative of the background supplemented with regional anthropogenic emission sources, considering the tropospheric lifetime of 1-2 months (e.g. Williams et al., 2017). This results in the resident mean $CH_2O$ value being only a quarter of those seen for boreal summertime over Colorado. Under these conditions, the relative negative biases for $CH_2O$ increase to ~40-60% across the chemical modules, with IFS(CBA) having twice the negative bias compared to IFS(MOZ). Given the low biogenic precursors, the overall negative bias suggests a deficit in the chemical production term of $CH_2O$ likely from the limited oxidation of other VOC's or peroxy-radical termination reactions, combined with missing direct HCHO emissions (Green et al., 2021). For springtime, relative biases are higher than in boreal summertime for Colorado across chemical modules, with values for the Southern US being simulated with low relative biases of around 20%.

**4.3 Tropospheric NO₂**



**Figure 11**. The horizontal mean cross-section of NO₂ below 1km over the US domain for JJA 2014 (top) and DJF (2014/2015).



The seasonal horizontal mean distributions of tropospheric $NO_2$ for IFS(CBA) for JJA and DJF are shown in the left panels of Figure 11. The large spatial variability in $NO_x$ emissions can be clearly seen resulting in much more distinct regional differences compared to the corresponding plot for CO (Fig 7) for both seasons. For JJA, higher (near-)surface values occur on the Eastern and Western seaboards, as well as Colorado and Texas, associated with urban conurbations. Comparing differences shows that the export of $NO_2$ out of the source regions is more effective

in IFS(MOC) associated with the conversion of a large regional fraction of $NO_2$ into PAN under conditions of high VOC fluxes (see Sect 5; Fig S7; Fischer et al., 2014). Thus, increases of $> 40\%$ occur in the surrounding oceans as a result of long-range transport of $NO_x$ out of the continent.

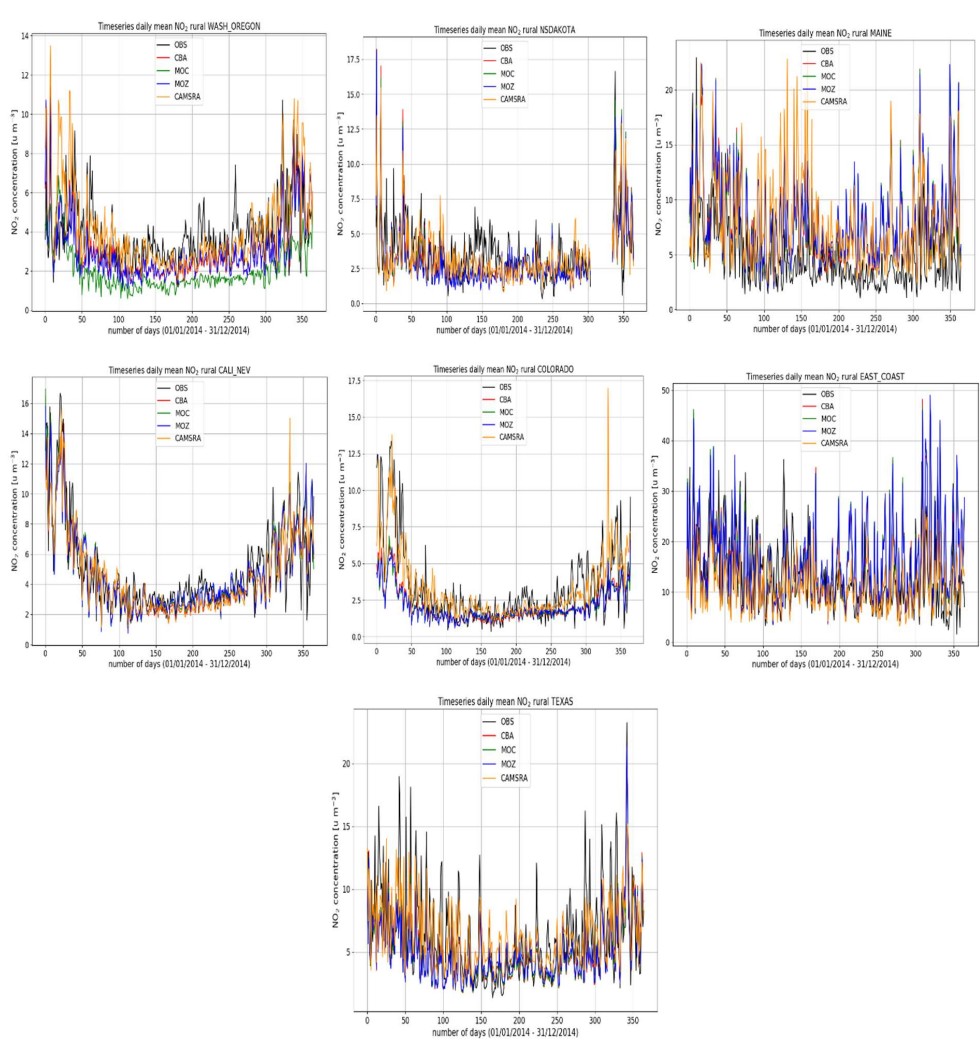


**Figure 12**. As for Fig. 5 except for surface $NO_2$.

A contributory factor to the simulated differences is the application of the new recommendation for $HNO_3$ formation which results in a ~10% lower gas-phase formation rate in IFS(CBA) (Stavrakou et al., 2013) under identical OH availability. The higher regional OH (c.f. Fig. S1) results in a stronger termination flux into $HNO_3$,

which is typically over-estimated in the region (c.f. Fig S6). Colorado is the only region where a negative offset occurs with respect to IFS(CBA), indicating locally higher conversion of $NO_2$ into other $NO_y$ components due to high regional VOC fluxes. For IFS(MOZ) lower $NO_2$ mixing ratios occur in the North US, with less export and



high mixing ratios in the Southern US. Some of these differences exist due to differences in the flux of the NO + $HO_2$ recycling term between chemical modules as a result of a difference in the $HO_x$ chemistry (c.f. Fig S1).

The associated seasonal mean distribution for NO is shown in the supplement (Fig S2), where maximal mixing ratios (0.3-0.4 ppb) occur over the more polluted urban areas. Substantially higher NO is simulated in the LT in IFS(MOC) and IFS(MOZ) compared to IFS(CBA), with continental increases of between 40-50%. Higher NO increases the direct titration term for $O_3$, with IFS(MOZ) having the lowest biases in $O_3$ for boreal summertime as diagnosed in the aircraft campaigns (Table S1). Moreover, the higher OH for JJA in both IFS(MOC) and
IFS(MOZ) (c.f. Fig S1) also increases direct gas-phase conversion of $NO_2$ into $HNO_3$ (c.f. Fig S5). Heterogeneous conversion of $N_2O_5$ to $HNO_3$ on wet surfaces and particles during nighttime is another important pathway for reducing $NO_x$ recycling, as analyzed in more detail in the next section.

    In figure 12 we make comparisons of the daily mean variability in surface $NO_2$ in the simulations against daily mean composites assembled from the AirNow network for rural stations within the regions defined in Table 10.
The number of stations used for assembling the composite is lower (higher) than that used for surface $O_3$ (CO) and typically not measured at the same location (c.f. Fig. 1 and Table 10). The uncertainty in the observations is higher than for the other longer-lived species and dependent on the instrumentation used in the local networks, especially for the lower concentrations. Except for the East Coast, Maine and Colorado, there is a notable annual cycle, with minima occurring during boreal summertime which is captured across the simulations. Mean daily biases for the
various chemical modules are region specific showing the influence of the chemical mechanisms across different chemical regimes. For boreal wintertime, negative mean daily biases of between 0-30 μg m$^{-3}$ occur throughout the US except for Colorado, where there are lower negative biases of between 0-10 μg m$^{-3}$, associated with high $NO_x$ and VOC emissions (c.f. surface $O_3$ discussed above). For Colorado, CAMSRA has a significantly lower mean daily bias of up to 2 μg m$^{-3}$ under different emission estimates and the use of assimilation. The corresponding
correlation ranges between R=0-0.5 for all of the simulations across domains, revealing only a weak-to-moderate correlation in the simulated fields. For boreal summertime the mean daily bias is typically negative between 5-30 μg m$^{-3}$ with only a limited difference between the three chemical modules, with CAMSRA exhibiting strong similarities. The extent of correlation is somewhat station specific with no latitudinal or longitudinal influence, typically ranging with R=0±0.2, thus weakly correlated throughout the stations. For CAMSRA, there is more anti-
correlation (R ranging from -0.1 to -0.3). For the more polluted urban stations, the seasonal cycle is captured well with observed mixing ratios ranging from 30-40 μg m$^{-3}$ during boreal wintertime to 7-15 μg m$^{-3}$ during boreal summertime, thus approximately twice those measured for the rural stations (not shown). Biases are somewhat higher for the winter months than shown for the rural comparisons (10-15 μg m$^{-3}$), compared to summer months (<10 μg m$^{-3}$). Therefore the differences in the biases between rural and urban environments is not as large as for
CO (not shown).

    As for other species, tropospheric $NO_2$ profiles are compared against the corresponding aircraft composites for the various campaigns in Figure 13, with the quantification of the biases given in Table S5. The corresponding profiles for tropospheric NO are shown in the Supplementary Material (Figure S2 and Table S4). As for the other trace gases, the shape is captured relatively well, with the profile exhibiting a negative gradient with respect to pressure.
Comparing the observational mean values in Table S4 shows that both Colorado and the East Coast have similar environments for $NO_x$ levels (around 1.0-2.5 ppb). Model biases increase from summertime to wintertime under such $NO_x$ rich conditions. For IFS(CBA) there is no significant change in the biases for many months when compared to CAMSRA. In the various aircraft composites for Colorado, the mini-ensemble shows a significant peak around ~820-830 hPa that is typically not seen in the observations and can result in positive bias in the LT.
For NO, the corresponding biases are lower than for $NO_2$ with only marginal differences between chemical modules. This suggests that differences shown in Fig S2 are driven by nighttime chemistry which is outside the observational sampling window for most aircraft campaigns. For springtime, for Colorado and Texas there are significant positive biases for both NO and $NO_2$, especially for IFS(CBA) which over-estimates by ~50%.



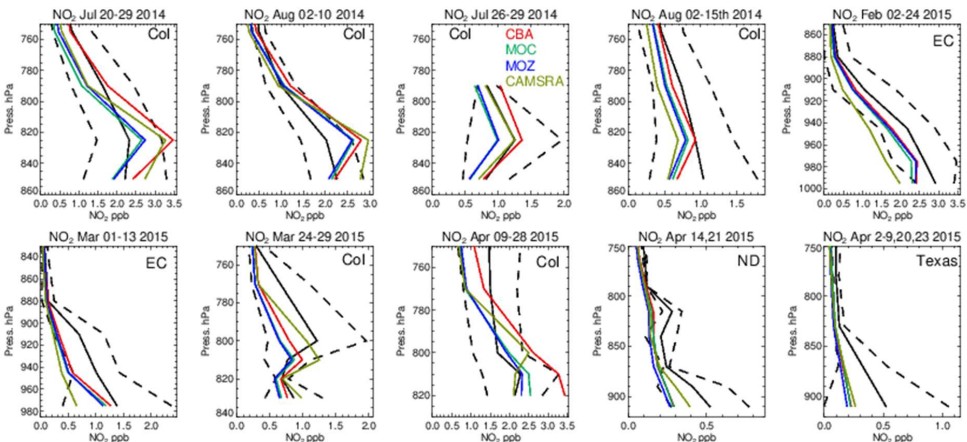

**Figure 13**. As for Fig. 6 except for tropospheric NO$_2$. Campaigns shown (top left to bottom right) are DISCOVER-AQ, FRAPPE, WINTER and SONGNEX.

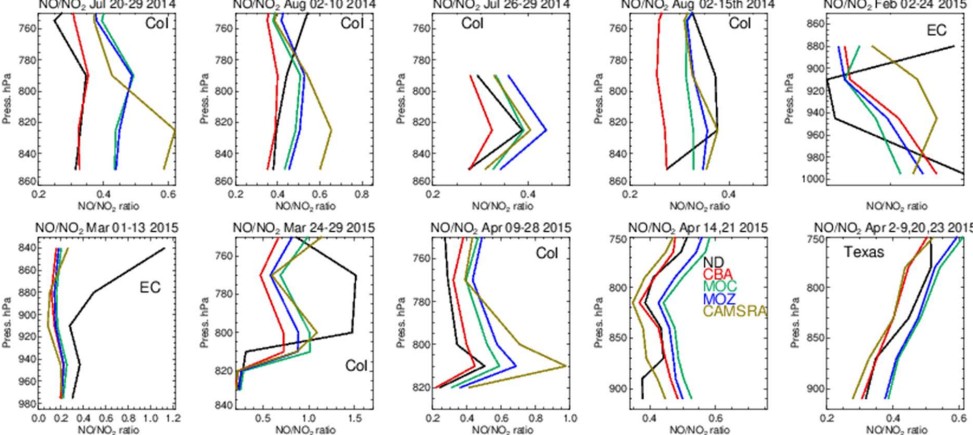

**Figure 14**. Comparisons of the NO/NO$_2$ ratio for the lower troposphere from the various aircraft campaigns, with observational values being shown in black. Campaigns shown (top left to bottom right) are DISCOVER-AQ, FRAPPE, WINTER and SONGNEX.

The NO/NO$_2$ ratio ($R$) is an indication of the equilibrium position of the NO$_x$ chemical system, as determined by the balance between fast titration of O$_3$ by NO plus conversion by peroxy-radicals, and its photochemical production from NO$_2$ photolysis. Low values for $R$ are indicative of an equilibrium which favours O$_3$ production and high values correspond to an equilibrium which favours O$_3$ destruction. A comparison of $R$ between the different chemistry modules and those derived from in-situ aircraft observations for the chosen campaigns are shown in Figure 14. For summertime over Colorado the variability in $R$ across the various campaigns ranges from 0.3-0.6, with significant differences across chemical modules. The profile shapes of $R$ in the LT are captured fairly well, although with lower variability in the chemical modules than in the observations. IFS(CBA) captures the correct $R$ value in the LT for many of the months whilst both other chemistry modules have a higher $R$, which moderates the O$_3$ biases shown in Table S1. Also, CAMSRA has higher $R$ values typically overestimating resident NO mixing ratios (c.f. Table S5), despite the application of data assimilation for NO$_2$. For the wintertime near the East Coast, $R$ is influenced by both daytime (high $R$) and nighttime (low $R$) measurements for February and mostly nighttime measurements for March, resulting in diverse behaviour between the two months shown. Model biases of ~0.1-0.2 occur across chemistry modules near the boundary layer. Interestingly there is a significant increase in


$R$ in the measurements once in the Free Troposphere (> 1.0) which is not captured by any of the chemical modules, which is associated with an underestimation of NO in the model.

For springtime over Colorado there is marked difference in profile shape compared to the summertime, with much more vertical variability showing differences in transport and mixing patterns for this season not captured by the simulations. For other regions performance is better with a very good agreement for Texas showing that for the
correct conditions (chemical regime and transport) the chemical modules can capture a realistic description of the chemical system.

**5 Tropospheric NOy**

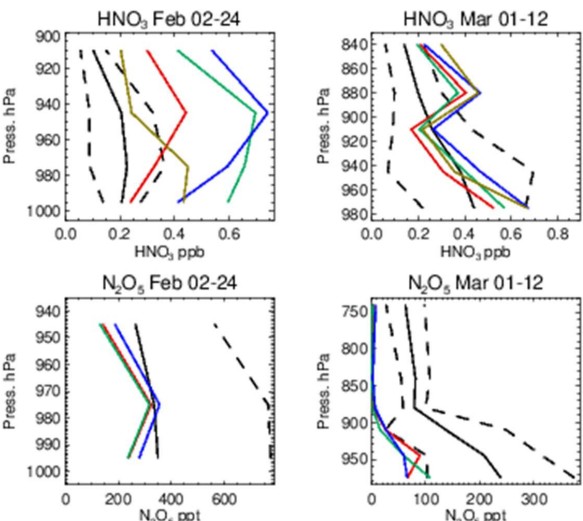

**Figure 15**. Comparisons of $N_2O_5$ and $HNO_3$ for the WINTER campaign using nighttime flights. The black line represents the observational mean value with the associated standard deviation being shown as the dashed line.

The formation of $O_3$ is determined by the resident $NO_x$ mixing ratios and the chain-length in the chemical recycling between NO and $NO_2$ before termination to $HNO_3$ occurs. One rapid loss route for $HNO_3$ production is the conversion of nitrogen pentoxide ($N_2O_5$) on wet surfaces and aerosols, which has been directly observed during
the WINTER campaign (Kenagy et al., 2018). For this campaign, $N_2O_5$ hydrolysis was shown to account for 58% of the chemical loss of $NO_x$ (Jaegle et al., 2018), with reaction of $NO_2$ with OH accounting for another third (thus the two most dominant chemical processes). In-situ observations of $N_2O_5$ are rare, where daytime mixing ratios are typically of the order of tens of ppt due to the efficient loss by photolysis, and, therefore, subject to high measurement uncertainty. During nighttime, surface observations range from 50-3000 ppt, with high daily
variability (e.g. Wood et al., 2005; Brown et al., 2009). The WINTER campaign includes observations taken during night, where accumulations of $N_2O_5$ occur, allowing model evaluations to be made.

Evaluations against the nighttime $N_2O_5$ measurements from the WINTER campaign are shown in Figure 15, along with the corresponding profiles of $HNO_3$. Unfortunately, no model data was available for $N_2O_5$ in the CAMSRA dataset for comparison. The formation of $N_2O_5$ involves the $NO_3$ radical, principally formed by the slow oxidation
process from the reaction of $NO_2$ with $O_3$ during nighttime. Their relatively small biases for the WINTER campaign (c.f. Fig 6 and Fig 13) provide some confidence in the flux of $NO_3$ production, where different reaction kinetics for thermal equilibrium are applied in each chemical module. Little difference exists across the chemistry modules for the simulated mixing ratios of $N_2O_5$, although a signature does exist for February regarding IFS(MOZ), which has higher mixing ratios in the LT by 10-20%. In this chemistry module no $N_2O_5$ conversion
on cloud particles and ice droplets is assumed.



For February a positive bias exists across chemical modules with respect to HNO$_3$ observations, where, counterintuitively, IFS(MOZ) has higher mixing ratios and biases (similar to IFS(MOC)) in spite of less efficient heterogeneous conversion. The corresponding N$_2$O$_5$ profiles indicate strong negative biases thus suggesting too rapid hydrolysis into HNO$_3$. As described in Sec. 2.2, the conversion rate is computed in the IFS from the available

Surface Area Density (SAD) of clouds and aquated aerosols, and the conversion frequency γ on these particles, (Brown et al., 2009), which is here assumed in the range of 0.01-0.02 depending on aerosol type, see Table 6. Derivations of γ(N$_2$O$_5$) from a chemical box modelling study based on the measurements across regions and scenarios taken during the WINTER campaign found a median value of 0.0143, with a spread of two orders of magnitude (~0.001-0.1, McDuffie et al., 2018). This suggests that to reconcile the negative N$_2$O$_5$ model bias

shown here the adopted γ(N$_2$O$_5$), needs to be made more variable. The inclusion of N$_2$O$_5$ conversion on clouds and ice particles for IFS(MOZ) would not improve on the biases shown here, without modification of the efficiency of other N$_2$O$_5$ loss routes.

Comparing the [HNO$_3$]/[N$_2$O$_5$] ratios to assess the efficiency of conversion whilst screening for very low and high values, Figure S10 shows that the [HNO$_3$]/[N$_2$O$_5$] derived from the chemical modules are only weakly correlated

with those observed ($R$=0.2-0.5), with the range of observational values being typically an order of magnitude lower than those seen in the mini-ensemble. For IFS(MOZ) during February there is a negative $R$ value possibly linked to the lack of N$_2$O$_5$ conversion on clouds, with IFS(CBA) and IFS(MOC) having positive $R$ values albeit with binned values which exceed 50 (too efficient conversion). For March IFS(MOZ) has the highest correlation and lowest mean bias which suggests conversion on aerosol surfaces dominates for this month and region. Table

S4 shows that NO$_2$ has a small negative bias for these months, although no validation of the nitrate radical (NO$_3$) can be performed to determine whether the deficit in N$_2$O$_5$ is only due to heterogeneous processes. IFS(MOC) exhibits the most occurrences of high ratios for February (> 20), whilst for March both IFS(MOC) and IFS(CBA) have a similar incidence of high ratios (> 50).

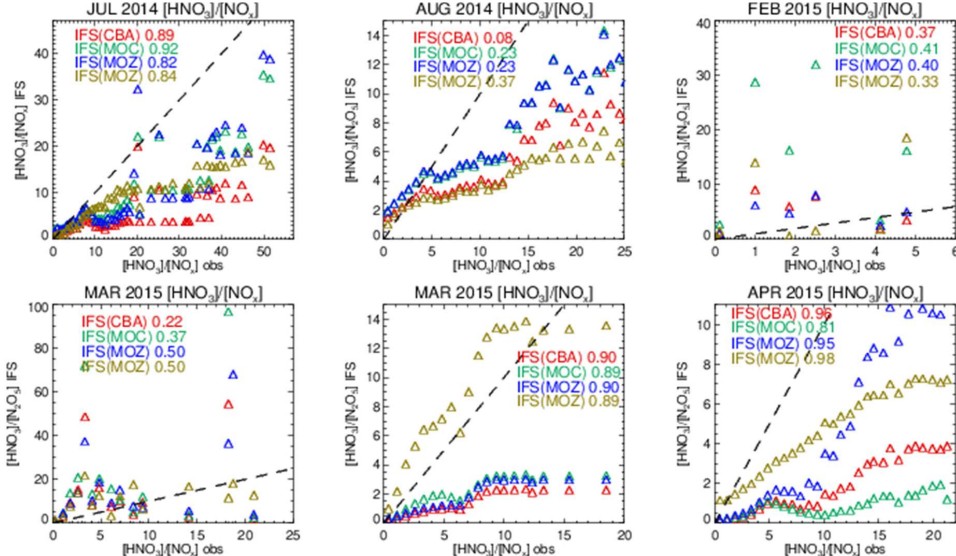

**Figure 16**. Comparisons of the observed [HNO$_3$]/[NO$_x$] ratio for selected months against the corresponding values from the ensemble members below 800hPa. The values are binned with respect to the observations using a bin-width of 0.75 with respect to [HNO$_3$]/[NO$_x$] and the correlation coefficients provided in each panel. Measurements are taken from FRAPPE, WINTER and SONGNEX campaigns to cover seasonality and location.

1. The length of the NO$_x$-cycle can be determined by examining the ratio between resident HNO$_3$ mixing
ratios with the associated NO$_x$ mixing ratios (denoted as $R'$=HNO$_3$/NO$_x$), where high values are indicative of a large fraction of N that has been made inert (in the form of HNO$_3$). The resulting comparisons of $R'$ calculated by accumulating values from selected days for each month (as for the vertical profile comparisons) for pressure levels below 800 hPa are shown in Figure 16. A bin-width of



0.75 in *R'* is used to calculate mean values in each respective bin. The resulting Pearson's (*r*) correlation coefficients are given in each panel.

For the spring and summer months over Colorado there is a high degree of correlation between measured and modelled *R'*, with correlations in the range 0.8-0.9, with the exception of August 2014, where the simulations become uncorrelated. For low *R'* values there is a tight agreement between the mini-ensemble members and the observational values. For higher *R'* values (> 10) the chemical modules exhibit low biases under high $NO_x$
emissions (see Table S3 and Fig. S5), although with variable biases for $HNO_3$ (c.f. Fig S6). Previous derivations of *R'* have found values in the range 0.8-10.4 for low $NO_x$ environments (Huebert et al., 1990), which is in the range of those observed in the remote free-troposphere. Biases for CAMSRA are overall much lower than those from the three recent chemistry modules during springtime.

Both daytime and nighttime measurements are used for deriving the wintertime (February-March) correlations, where the observational range in *R'* is approximately half that derived for summertime. Correlations become much
weaker and the *R'* values are an order of magnitude higher for the mini-ensemble than those in the observations, indicating that the $NO_x$ chain-length for the chemistry versions is shorter than observed. One main difference is that uncertainties associated with heterogeneous conversion of $N_2O_5$ plays a dominant role in $HNO_3$ production during nighttime, which may explain the reduced correlation. For CAMSRA, the high *R'* values (> 30) during
February in the three chemistry modules, does not occur.

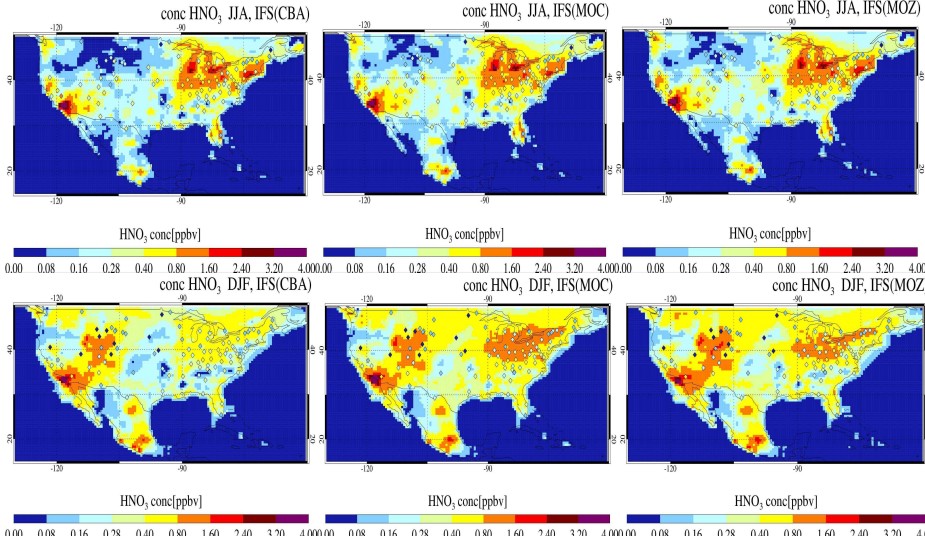

**Figure 17.** The Seasonal mean distribution of surface mixing ratios for gas phase $HNO_3$ across the US for (top) JJA and (bottom) DJF against seasonal composites of observations taken from the CASTNET network. The
observational mean values and locations of the measurement stations are given as the diamonds located in each panel.

In order to further evaluate the regional differences in $HNO_3$ across the chemical modules, we make seasonal comparisons of surface $HNO_3$ mixing ratios against observational composites taken from the CASTNET network throughout the US, Figure 17. The maximal mixing ratios for $HNO_3$ occur near regions with high $NO_x$ emissions.
For JJA differences between configurations are modest, with the largest percentual spread over the comparatively clean northern part. Comparing seasons shows that for IFS(CBA) during DJF much more $HNO_3$ exists towards the western US than for JJA. Instead, there is a significant reduction in IFS(CBA) during DJF on the East Coast that is larger than for the other two modules.

Despite relatively small absolute values in OH mixing ratios during DJF, the significant percentual differences
between modules (Fig S1), could be responsible for differences in the direct production term across chemical



modules, with overall higher HNO₃ mixing ratios in IFS(MOZ) and IFS(MOC) than IFS(CBA). Also, there are uncertainties in HNO₃ loss through nitrate aerosol formation and deposition contribute to uncertainties in HNO₃ burden, e.g. Nowak et al. (2010).

Spatial correlations $r$ between the seasonal mean modeled and observed mixing ratios for the Eastern and Western US are approximately $r$=0.5 for each of the chemistry modules for JJA. For this season the chemistry modules show an associated mean positive bias of between 0.3-0.4 ppb for the Eastern US and almost no significant mean bias for the Western US, due to some cancellation of positives and negatives for different locations.

The corresponding seasonal mean values for DJF show more divergence in correlation statistics with $r$ ranging from 0.07-0.24 for the Eastern US to 0.54-0.65 for the Western US for the three chemistry modules. Associated
mean biases are in the range 0.17-0.43 ppb (Eastern US) and 0.41-0.51 ppb (Western US), respectively. These evaluations indicate significant uncertainty regarding surface HNO₃ model capabilities, with overall positive biases. We refrain from further analysis of HNO₃, as this involves assessment of deposition, and nitrate formation. Finally, it should be acknowledged that the CASTNET measurement network has been shown to have rather low mixing ratios when compared to alternative measurement networks (Lavery et al., 2009).

The significant differences in the seasonal distribution shown for NO₂ as diagnosed in Sec 4.3 can, in part, be explained by variability in PAN production and loss across the various mini-ensemble members. A particularly large model spread was seen for DJF (Figure S7). Here colder temperatures increase its tropospheric lifetime by suppressing thermal decomposition, but simultaneously decrease its formation in absence of biogenic and BB precursor emissions, Fischer et al. (2014).

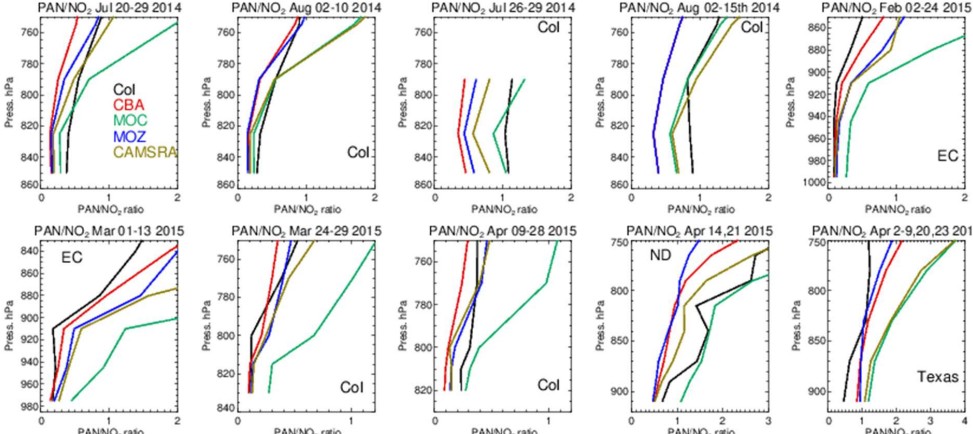


**Figure 18**: Comparisons of the ratio of PAN/NO₂ derived from the corresponding mean values derived during each aircraft campaign. Campaigns shown (top left to bottom right) are DISCOVER-AQ, FRAPPE, WINTER and SONGNEX.

Figure S8 compares resident mixing ratios of PAN against composites for the different aircraft campaigns, as those
used to evaluate NO₂. It shows that for JJA, IFS(CBA) and CAMSRA typically under-estimate PAN mixing ratios in the LT, where biases decrease as the temperatures get colder for the wintertime. For December-March, IFS(MOC) shows positive biases of up to 100% indicating that PAN is too stable under cold conditions. This points at different reaction data employed compared to IFS(CBA) and IFS(MOZ), and a lower photolysis frequency. As a result, a significant overestimation in the fraction of NOₓ exported out of the source regions will
occur, as shown in the seasonal zonal mean PAN distributions in Figure S9, where twice as much northerly transport occurs for IFS(MOC) compared to the other chemistry modules.

The ratio $F$ defined as $F$=PAN/NO₂ can be used to examine the ability of the chemical modules towards capturing the correct partitioning of resident NOₓ into PAN, which can then be transported out the source regions by convective uplift and long-range transport affecting background O₃ budgets (Fischer et al., 2014). In general, the



observations show an increase in $F$ with respect to altitude, with $F$ typically ranging between 0-1 during summertime and 0-0.2 during wintertime, Figure 18. For most months and regions, IFS(CBA) and IFS(MOZ) provide accurate simulations of the vertical variability in $F$ values below 850 hPa. IFS(MOC) generally has a positive bias in $F$, particularly during wintertime, where F is up to a factor 2 higher than that observed. This is indicative for a too stable PAN in this chemistry version, which affects the $O_3$ production efficiency via the
availability and distribution of $NO_2$.

During summertime over Colorado IFS(MOC) exhibits good agreement in $F$ in the boundary layer, with IFS(CBA) and IFS(MOZ) under-estimating by 0.03-0.05. In spite of the updates of the $NO_x$ chemistry in IFS(CBA), CAMSRA has slightly lower biases for summertime.  For wintertime (springtime) towards the East Coast (Colorado) $F$ ratios for IFS(MOC) are nearly double those observed. However, agreement is quite good for
North/South Dakota for April showing the regional variability in performance of the chemical modules. As PAN is transported out of the boundary layer the contribution of the loss rate due to photolysis increases (albeit with a low frequency thus allowing long transport lifetimes). This highlights the importance of a correct parameterization of the photolysis frequency across the various chemistry modules. The reaction towards the colder temperatures in the FT markedly affects lifetime, where different reaction kinetics are applied across chemical modules. On
average an improvement for IFS(CBA) is seen with respect to CAMSRA, which can be attributed to improvements in model $NO_y$ chemistry when considering the corresponding negative bias for $NO_2$ shown in Figure 13.

### 6 Conclusions

In this study we have presented a detailed description of the recent updates which have been made to the chemistry
modules that are integrated in ECMWFs IFS global model for the purpose of performing global Air Quality forecasts. We have evaluated a set of three simulations covering the years 2014/2015, using the latest model configuration as developed in an experimental version of the ECMWF IFS cycle 47R1. This provides insight in the performance of the modeling of trace gases (here excluding data assimilation) in the CAMS global system. This study has focused on lower tropospheric composition for the contiguous United States with an emphasis on
tropospheric $O_3$, $NO_2$ and CO. We also included comparisons against the most recent reanalysis dataset that is based on a previous CAMS configuration (CAMSRA). This allows assessment of model changes compared to this established dataset.

By comparing seasonal means in the lower troposphere between the various chemistry modules we have shown a strong seasonality in the regional inter-model differences for $O_3$, CO and $NO_2$ in the US. For $O_3$ these differences
are limited to ±5% during boreal summertime, during which higher mixing ratios occur. The ability to capture the regional seasonality in surface concentrations for the background is somewhat region dependent, with relatively good agreement for the West Coast and an overestimation towards the East Coast. Comparing seasonal composites against ozonesondes shows that there is generally good agreement in more remote locations and high positive biases of 10-30 ppb for more polluted regions, especially at the surface near the US East Coast. Comparisons for
more southerly regions show lower mean daily biases in Texas and California-Nevada with limited correlation in the daily variability. For the Colorado region, there are biases of ±6 ppb across chemistry modules (±10-15%). At the surface there are small negative biases of around 5 $\mu$gm$^{-3}$ for IFS(CB05BASCOE) and 15 $\mu$gm$^{-3}$ for CAMSRA. For boreal wintertime a significant variability in the $O_3$ production efficiency occurs across chemistry modules resulting in IFS(MOCAGE) and IFS(MOZART) exhibiting increases in mixing ratios of +6-15% and +20-25%
across a wide region as compared to IFS(CB05BASCOE), especially in the Northern US. A significant positive bias in surface concentrations occurs for 2014 in the Northern US indicating too efficient $O_3$ production, whereas CAMSRA exhibits a significant negative bias. Other regions show less difference across the simulations.

Associated differences occur for the OH radical for both seasons, which leads to significant differences in the tropospheric distribution of CO of between 8-20%, especially during wintertime. In general, the seasonal cycle at
the surface is captured well when compared to both ESRL background observations and surface AirNow, with IFS(MOZART) exhibiting the largest negative biases in Northern US regions. When compared against aircraft observations positive biases in CO of 10-20 ppb occur for the Colorado region during boreal summertime for IFS(CB05BASCOE), with IFS(MOCAGE) agreeing relatively well and IFS(MOZART) underestimating by a few ppb. These biases turn negative for wintertime and spring, reaching underestimates of upto 10-35 ppb across the
simulations for different regions and months. This leads to deficits of 30-35 ppb in the lower troposphere for IFS(MOZART). Analysing similar aircraft comparisons for $CH_2O$ for both seasons shows negative biases of $CH_2O$



between 5-40% depending on the region, which contributes to the negative CO biases. Biases of CO for all chemical modules are typically larger than the CAMSRA dataset which is strongly constrained by assimilation of CO observations from satellite retrievals.

As was the case for $O_3$, also $NO_x$ shows a seasonal variation in the simulated inter-model differences within the order of 5-10% for $NO_2$ and up to 50% for NO. Comparing profiles for both trace gases against aircraft measurements shows significant negative biases exist for both NO and $NO_2$ for the $NO_x$-rich environment of Colorado across all chemical modules indicating regional emissions which are too low. The performance of the three chemistry versions is overall better than for CAMSRA, which can be understood by the fact of the limited

impact of $NO_2$ data assimilation in the CAMS reanalysis. Comparisons against AirNow surface observations shows that the regional annual cycles are captured well across the simulations with negative biases and showing only a weakly correlated daily variability.

Examining $NO/NO_2$ ratios shows that the equilibrium between NO and $NO_2$ is mostly captured well by IFS(CB05BASCOE) in the boundary layer, with the other chemical modules overestimating the fraction of NO

(albeit with lower $NO_x$ mixing ratios). A strong correlation exists in the $HNO_3/[NO + NO_2]$ ratio across days for boreal summertime between the modelled and measured fields ( R> 0.9), albeit with a negative model bias of ~50%. This is indicative of a lower $NO_y$ burden in the simulations due to cumulative differences in emissions, chemistry, aerosol formation and deposition processes. For CAMSRA, the $HNO_3/[NO + NO_2]$ ratio is overall better. For nighttime under cold conditions, the $NO/NO_2$ ratio is typically underestimated implying a lack of NO

regeneration by slow redox reactions.

There is generally an overestimation in $HNO_3$, both at surface and in the free troposphere, which may be due to too efficient $N_2O_5$ hydrolysis on wet surfaces under some conditions. Model analysis suggests that this conversion on cloud surfaces is not a dominating term with respect to associated $N_2O_5$ comparisons for the East Coast during the wintertime period.

One dominating factor on the seasonal distributions of $NO_2$ is the fraction stored as PAN and transported out the source regions. For boreal summertime, IFS(MOZART) simulates 20-50% less resident PAN than IFS(CB05BASCOE), which contributes to the more efficient $O_3$ formation in IFS(MOZART). When comparing against aircraft profiles around Colorado for July and August, there is generally an underestimation in resident PAN of 40-60% across chemical modules, suggesting a lack of Volatile Organic Compounds precursors and

subsequent acetyl-peroxy radicals, in line with previous studies (Huijnen et al., 2019). For boreal wintertime, when there is an extended tropospheric lifetime under cold temperatures, significant positive biases in regional PAN were diagnosed as compared against aircraft profiles for IFS(MOCAGE), pointing at differences in model assumptions regarding the stability of PAN, as determined by the rate data employed. This is also reflected by the $PAN/NO_x$ ratios which show a strong overestimate in IFS(MOCAGE) and requires future developments.

As presented in this manuscript, a significant divergence of key air quality products simulated by each of the chemistry modules exists, depending on seasonal and regional conditions. These are due to fundamental differences associated with the oxidative capacity and the regional efficiency for the production of tropospheric $O_3$, which are in turn determined by the chemical mechanism, the parametrizations adopted and the rate data used. In future studies attention should be made towards (i) improvement of variability in surface $O_3$, CO and $NO_2$, with

respect to air quality observations, by a joint effort of improving the emissions and deposition handling, and improved diagnostics (ii) further homogenization of the physical conversion processes across modules with respect to radicals and $N_2O_5$, (iii) improve on the VOC tropospheric burdens to provide sufficient peroxy-radicals for better PAN formation and (iv) further investigate what determines PAN mixing ratios under cold/low light conditions in term of dissociation and stability.

Whilst analysis of the three chemistry modules in CAMS provide a strong handle on uncertainties associated with chemistry modeling, the further improvement of operational products additionally requires coordinated development involving emissions handling, chemistry and aerosol modeling, complemented with data-assimilation efforts.

*Author Contributions*



JEW and VH were principal authors of the paper and conducted most of the evaluation against observational datasets. VH, IB, SP, BJ and VM performed the three individual simulations which were used for the evaluation. MM and TS helped with the evaluation against the AirNow surface observations. JF contributed to the interpretation of results.

*Code availability*

Model codes developed at ECMWF are the intellectual property of ECMWF and its member states, and therefore the IFS code is not publicly available. ECMWF member-state weather services and their approved partners will get access granted. Access to a version of the IFS (OpenIFS) that includes this experimental cycle may be obtained from ECMWF under an OpenIFS licence. More details at https://confluence.ecmwf.int/display/OIFS/OpenIFS+Home (last access: 28 Sept 2021)

*Data availability*

The CAMS reanalyses data are freely available from https://atmosphere.copernicus.eu/ (last access: 28 Sept 2021).

*Competing Interests*

The authors declare that they have no conflict of interest

*Acknowledgements*

We acknowledge funding from the Copernicus Atmosphere Monitoring Service (CAMS) under contract CAMS_42_II, which is funded by the European Union's Copernicus Programme. We acknowledge the World Ozone and Ultraviolet Radiation Data Centre (WOUDC) for providing ozone-sonde observations and the Earth System Research Laboratories network for providing the surface measurements of carbon monoxide and ozone. We acknowledge the AirNow monitoring network for allowing access to surface observational data for carbon monoxide, nitrogen dioxide and ozone. Finally, we acknowledge NOAA for access to the multiple aircraft campaigns used in this manuscript.



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
