# Peer review of "Regional evaluation of the performance of the global CAMS chemical modeling system over the United States (IFS cycle 47r1)"

_Geoscientific Model Development, 2021_

## Referee Comment (RC1)

Review of "*Regional evaluation of the performance of the global CAMS chemical modeling system over the United States (IFS cycle 47r1)*" submitted by Jason E. Williams et al. to Geoscientific Model Development

**Summary:** In this manuscript, the authors present a thorough intercomparison of coupled chemistry and meteorology model simulations against a suite of ground-based, balloon, and aircraft measurements of trace gases in order to assess the chemical updates made to three chemistry modules used in the ECMWF's IFS.  Here, the IFS version tested is a more modern IFS cycle 47r1 than used in the CAMS reanalysis (also included in the study as a baseline) and the chemistry modules are the CB05BASCOE, MOZART and MOCAGE and include tropospheric and stratospheric chemistry, although the focus of this manuscript is on air quality relevant species and their concentrations in the troposphere.  The authors draw conclusions regarding improvements and degradation of atmospheric composition with the new implementations during winter and summer seasons by comparing model output to the measurements of ozone, carbon monoxide, and nitrogen dioxide, and then further dive into the chemistry driving the changes using the additional chemical measurements from the campaigns and ground-based networks.  I outline below my major concerns regarding the manuscript, and therefore recommend this manuscript undergoes major revisions before it is published.

**Major Comments:**

The manuscript is very long, with 9 tables and 18 figures within the main text. There are 6 tables and 10 figures in the supplemental, which are often referenced in the main text in a manor that should be included as "results" and not just available to the reader if they are curious to learn more.  The authors need to take time to restructure the manuscript to bring out of the supplemental the tables and figures that are used in the paper for scientific discussion (Sections 4 and 5) and the conclusions (Section 6).  This may require being more creative in the presentation of the figures.  If there isn't new information, not all profiles or timeseries need to be shown; for example, instead make selections of only the ones that drive home the point the authors are trying to make, or maybe some of the months for a campaign can be combined.

I could not discern a clear metric for how the authors were measuring the improvements.  Often there were statements that the differences were "significant" or "agreed well" that are subjective not quantitative.  Is the main goal of the paper to identify if a configuration is suitable for the next version of CAMS?  If the IFS configuration can only be run by ECMWF and its partners, what is the impact of this paper for the community?

As I am based in the US now, I have observed the increased use of CAMS forecasts in the US, such as the inclusion in The Weather Channel for air quality forecasts.  The reason this study focused on evaluating the updated chemistry in their three chemistry modules used in a global model produced by the European Center over the region of the USA is not strong (because it is a polluted region and most evaluation has been done on more background environments) and

completely lacking from the abstract.  Europe and Asia are also "affected by high anthropogenic emission sources" and could have been studied.  If, for example, the US was chosen because of the vast amount of free, publicly-available observations from state and national ground-based and balloon-based networks and multiple field campaigns supported by national government agencies (NOAA and NASA), which the magnitude and variety of publicly-available observations cannot be matched by other regions/countries, including in Europe, state that.

As a scientist who runs global coupled chemistry and meteorology models, I take care to be considerate of other naming conventions around the world.  For a foreign author, I would let slide one or two mistakes regarding place names, but the constant errors are unacceptable and careless.  If unsure of a place name or a regional name, reach out to a contact who would be more familiar or from that country/region or spend some time searching through the literature to see how others have referenced it.  In particular, two regions in Figure 1 include several stations (and the Yarmouth ozonesonde location) that are in Canada, and Figure 17 shows a CASTNET station in Egbert, Ontario.  The authors may want to consider changing the paper to "North America" instead of "US".  In the technical comments, I indicate place names and regions in the paper that need to be revisited.

The manuscript has many technical errors; such a long manuscript requires careful attention to detail prior to submission.  Several acronyms are defined in multiple places or used before first defined. Unconventional use of variables and notation are mixed with conventional use; in particular I raise to your attention that the Pearson coefficient is defined as both R and r, as R is redefined to mean the NO to NO2 ratio.  This should be avoided.

**Technical edits:**
For the minor and technical edits, please see the highlighted text and 156 associated comments in the marked-up version of the manuscript below.

[Figure]

[Figure]

[revised manuscript text omitted]
$_3$COCHO + 0.125 HCOOH + 0.125 CH$_3$O$_2$ + 0.125 CH$_3$COOH + 0.05 CO | 2.0 x 10$^{-12}$ x exp(320/T) | [1] |
| HYAC + h$\nu$ → C$_2$O$_3$ + HO$_2$ + HCHO | As J(CH$_3$COCH$_3$) | [1] |

185 To date IFS(CBA) has only included a very simplistic parameterization for the oxidation of C$_5$H$_8$. To improve the realism of the product distribution from the oxidation of C$_5$H$_8$ by OH, we have developed a mechanism which is a hybrid of that developed by Stavrakou et al. (2010) and Lamarque et al. (2012). This new hybrid method includes the direct formation of glyoxal (CHOCHO), hydroxy-aldehydes (HPALD1, HPALD2), a peroxy-product (ISOPOOH), glycolaldehyde (GLYALD), hydroxy-acetone (HYAC) and methyl-glyoxal (CH$_3$COCHO). Explicit

190 *J* values are calculated online for these products using the latest recommendations for absorption data, as for other species. Most of these intermediates are soluble with Henry solubility analogous to ALD2 except CHOCHO, where the approach of Ip et al. (2009) is used. We still retain the ISPD intermediate representative of methyl-vinyl-ketone and methacrolein from previous versions of the scheme. Reaction rates in this mechanism have been updated following latest recommendations, as aligned with the mechanism described by Myriokefalitakis et al. (2020).

195 Validation of this updated component of the IFS(CBA) is beyond the scope of this study, but we have found that OH-recycling increases over forested regions with high biogenic emission fluxes is as for that shown in Stavrakou et al (2010) thus affecting atmospheric lifetime of individual trace gases for regions with high resident mixing ratios of C$_5$H$_8$. It should be noted that for O$_3$ and NO$_3$ we still adopt the original stoichiometry in product distribution as in previous versions of IFS(CBA) (Huijnen et al., 2016). We provide details related to this extensive

200 update in Table 3.

To date aromatics were not explicitly treated in IFS(CBA), but rather as part of the generic paraffinic bond tracer PAR. This is now updated. For this we follow the work of Karl et al. (2009), who describe the oxidation of the aromatic tracers toluene (TOL) and xylene (XYL), allowing a coupling to Secondary Organic Aerosol (SOA) formation. In addition, in our model version the product distributions and rate expressions for NO and HO$_2$ radical-

205 radical reactions are taken from Myriokefalitakis et al. (2020). This links the aromatic species towards oxidant loss (O$_3$, OH, NO$_3$), the production of CHOCHO/CH$_3$COCHO and allows the introduction of gas-phase precursors for SOA formation (SOG) from anthropogenic and biomass burning sources. We provide details on the extension to the aromatic chemistry in IFS(CBA) in Table 4.

**Table 4**: Updates to the oxidation of TOL and XYL as implemented in the IFS(CBA) chemistry as compared to
210 Huijnen et al. (2019). The reaction scheme is adapted from Karl et al. (2009), with modification to the product distribution for loss of AROO$_2$ following Myriokefalitakis et al. (2020). (*) This indicates the final rate applied accounts for the ortho-, meta- and para-isomers of the cyclic aromatics.

[revised manuscript text omitted]

---

## Referee Comment (RC2)

**Review of Regional evaluation of the performance of the global CAMS chemical modeling system over the United States (IFS cycle 47r1) by Williams et al.**

**General summary**

This paper updated the homogeneous and heterogeneous NOx chemistry applied in the three independent tropospheric-stratospheric chemistry modules maintained within CAMS and evaluate lower tropospheric O3, CO, and NO2 over the contiguous US. The developments are evaluated against multi-platform observations and the chemically induced biases in different species is estimated to be within 10 ppbv. The authors find large model spread between chemistry schemes during winter in lower tropospheric ozone (10-35%) and oxidative capacity which is attributed to NOx lifetime differences. The study concludes that chemically induced differences in quality of CAMS forecasts over CONUS vary by season, species, altitude, and region. The intercomparison of different chemistry configurations is interesting but I find several major weaknesses in the paper as listed below.

First, I was expecting that the authors will evaluate the impact of the updates to the homogeneous and heterogeneous chemistry against the older model version without these updates to demonstrate how important these updates here. However, such a comparison is not presented. Thus, the real value of these updates is not quantified.

Second, the authors compared the three chemistry configurations against the CAMS reanalysis (CAMSRA). At lines 76-77, the authors state that any changes in the operational system should ensure that the change improves the model performance with respect to observations. In most cases (e.g., Figures 4-6), the performance of CAMSRA is better than all the three individual configurations. Since any of the new configurations here do not outperform CAMSRA, does it mean that these configurations may not be suitable for operational adaptation? Furthermore, the assimilation of several species in CAMSRA makes a direct comparison of three chemistry simulations with CAMSRA difficult.

Third, several discrepancies between the observations and model simulations are not clearly explained (see specific comments for details).

Fourth, the choice of US as a polluted region over Asia is not motivating enough given that anthropogenic emissions in Asia are much higher than the US.

Therefore, I recommend major revisions of the paper before it can be considered for publication in GMD. Below are some specific comments.

**Specific comments**

Lines 57-58: Operational air quality forecasts over the US are not only provided by the CAMS system. The US has their National Air Quality Forecasting Capability (NAQFC) that provides air quality forecasts at 12 km grid spacing twice per day and the NAQFC forecasts are used by

decision-makers to issue advance notices and warnings for air pollution episodes. Since lines 43-55 discuss aspects and benefits of air quality forecasting, it is relevant to discuss NAQFC here.

Lines 63-64: Anthropogenic emissions in the US are much lower than Asia and thus it is hard to see this as a motivation for focusing on the US.

Line 99: To what heterogeneous reactions you are referring here?

Line 105: I suggest replacing the term "mini-ensemble" with "multiple chemistry simulations".

Line 144: Do all the chemistry configurations use the same quantum yield and cross sections for calculation of photolysis frequencies? Also, how are the effect of aerosols treated in photolysis rate calculation? If not, can the 5% differences in photolysis frequencies be attributed to these factors?

Tables 2-5: Please include a description of various symbols (Ko, R, T, and QY etc.) used in the rate expression column in the captions of these table.

Line 168: Do you provide HONO emissions to the model as well?

Tables 3-5: Since you are not validating the updated isoprene, aromatic, HCN, and CH3CN chemistry, I think Tables 3-5 can be moved to supplementary material.

Line 442-443: The minimum is not centered around Colorado but rather is spread over Oklahoma, Nebraska, South Dakota, North Dakota, and parts of Texas.

Line 446: Can you explain why the decrease in MOZ is due to reduced transport? What are the source regions affecting the surrounding oceans?

Fig S1: The differences in OH are interesting. While it makes sense that higher ozone can lead to higher OH levels through photolysis but I am unbale to understand why OH increase in summertime in MOZ over the western US despite of a decrease in ozone mixing ratios. Any thoughts?

Figure 4: The observed ozone seasonal cycle at NWR looks suspicious. Most of the sites in Colorado show a summertime maximum. I looked at multi-year hourly ozone time series from NOAA GML website (see below) and found that there are a lot of measurement gaps at NWR during 2014. So, I recommend revisiting the data processing and evaluation at this site and revising the corresponding text accordingly.

[Figure]

Figure 5: None of the model configurations capture ozone seasonal cycle at rural East Coast sites. All the models predict a summertime peak while the observations show continuous decrease. This is really surprising. Can you throw some light on this?

Lines 535-540: Here you use ppb as ozone units while all the earlier discussion is in micrograms/m3. Please use consistent units.

Line 581-582: Can you provide more insight into the different seasonal cycles of CO between the model and observations at NWR?

Lines 775-783: This is a good discussion but it would have been great to perform a couple of sensitivity experiments using just one of the chemistry configurations to understand the impact of conversion efficiency on N2O5 hydrolysis.

---

## Author Comment (AC1)

Response to Dr Knowland: We thank Dr. Knowland for her comprehensive review of our manuscript and for providing a detailed guide towards proposed changes. This has led to significant updates with respect to domain definitions, selection of figures, and discussion of quantitative differences between modules, in the updated version of the manuscript. We provide detailed responses to the points raised below.

*The manuscript is very long, with 9 tables and 18 figures within the main text. There are 6 tables and 10 figures in the supplemental, which are often referenced in the main text in a manner that should be included as "results" and not just available to the reader if they are curious to learn more. The authors need to take time to restructure the manuscript to bring out of the supplemental the tables and figures that are used in the paper for scientific discussion (Sections 4 and 5) and the conclusions (Section 6). This may require being more creative in the presentation of the figures. If there isn't new information, not all profiles or timeseries need to be shown; for example, instead make selections of only the ones that drive home the point the authors are trying to make, or maybe some of the months for a campaign can be combined.*

We agree that the manuscript is rather long, but this paper will act as a future reference with respect to the nitrogen cycle in the chemical forecast model which has not received such attention before and requires co-located high quality datasets to assess performance, necessitating an in-depth assessment. We do not feel moving everything to the main manuscript will improve the readability and want to show as many comparisons as possible rather than removing information that may prove useful for the community in the future. To improve on the repetition aspect, we remove panels from the $O_3$ and $NO_2$ AirNow surface comparisons whilst keeping some form of regional representativity, whilst including bias and correlation statistics for all of the US regions defined in new tables.

*I could not discern a clear metric for how the authors were measuring the improvements. Often there were statements that the differences were "significant" or "agreed well" that are subjective not quantitative. Is the main goal of the paper to identify if a configuration is suitable for the next version of CAMS?*

The main purpose of this study is to quantify the ability of the respective atmospheric chemistry modules to describe air quality forecasts for a region outside Europe affected by anthropogenic pollution, without the application of data assimilation. This study therefore provides a benchmarking of such a configuration and provides a quantification of the uncertainty due to the model chemistry employed. Moreover, analysing model performance outside the European domain, where the model has not been tuned, provides more confidence with respect to CAMS forecasts which are used by US partners. The wealth of freely available observational data for the US (ozonesondes, surface measurements, aircraft campaigns) allows quantification of model performance for this region.

In this study the CAMS ReAnalysis (CAMSRA) is used as a reference, as it is a widely used and relatively well-known dataset. Nevertheless, we realize that interpretation of differences with respect to CAMSRA cannot be attributed to model chemistry updates alone and should be considered with care. Finally, the configuration tested here, though interesting from a chemical point of view, will not be identical to the one used for (future) operations where the horizontal resolution used in this study is lower and data-assimilation of tropospheric species is not applied. A separate, operational activity will take place to define, test, verify, validate and move over to a next version of the operational system, taking into account not only changes to model chemistry, but also emissions, meteorology, transport and data assimilation. We have now added a sentence regarding the use of CAMSRA, thus : "However, attributing the differences in performance of chemical updates without data assimilation against CAMSRA is complicated by the simultaneous changes in IFS cycle and emission estimates since CAMSRA was produced." .

*As I am based in the US now, I have observed the increased use of CAMS forecasts in the US, such as the inclusion in The Weather Channel for air quality forecasts. The reason this study focused on evaluating the updated chemistry in their three chemistry modules used in a global model produced by the European Center over the region of the USA is not strong (because it is a polluted region and most evaluation has been done on more background environments) and completely lacking from the abstract. Europe and Asia are also "affected by high anthropogenic emission sources" and could have been studied. If, for example, the US was chosen because of the vast amount of free, publicly-available observations from state and national ground-based and balloon-based networks and multiple field campaigns supported by national government agencies (NOAA and NASA), which the*

*magnitude and variety of publicly-available observations cannot be matched by other regions/countries, including in Europe, state that.*

Whilst CAMS provides a regional forecast for air quality using an ensemble of regional models for Europe, for other regions over the globe CAMS relies on global model simulations to provide such information due to the size of the domains. Therefore, we argue that quantification of the model performance in terms of air quality aspects for another region than Europe, with specific emphasis on uncertainty due to atmospheric chemistry modeling, is of interest to the CAMS consortium and wider community. This is the first time that such an in-depth analysis of the CAMS system adopting independent chemical schemes has been performed for the US. We wanted to focus on a corresponding NOy analysis in order to explain the simulated tropospheric $O_3$ in the simulations, for which aircraft measurements are a unique source of collocated data allowing this analysis. No such measurements are available with such a frequency for either Europe or China.

*If the IFS configuration can only be run by ECMWF and its partners, what is the impact of this paper for the community?*

The reviewer raises a valid concern regarding the availability of the IFS configuration beyond ECMWF and its partners. In response, there is more nuance than suggested by the reviewer. Firstly, the products from the CAMS system are widely used, including users in the US. A critical assessment of the model against surface observation networks of the US has not been previously documented in such detail and is therefore of interest to the wider modelling community. Regarding the availability of the exact model version to the external community, this is indeed currently restricted. We provide a comprehensive document of the details of model simulations, which in principle allows those interested to model a similar configuration in terms of emissions, meteorology, and atmospheric chemistry.

Here we repeat, and provide further details on the availability of model code: the IFS meteorological model is available to research institutes as part of OpenIFS (currently cycle 43r3, i.e. the version also used for the CAMS Reanalysis, while an update to provide a more recent cycle is planned). The CB05 and CB05-BASCOE atmospheric model chemistry used in cycle 43r3 will be made available soon (Huijnen et al., in preparation, 2022), while updated versions of model chemistry (particularly as reported here), are expected to also become available along with future releases of OpenIFS itself.

In order to help the quantification of current model performance we have now included more bias statistics in the revised manuscript associated with the AirNow surface comparisons and make additional reference to the bias tables provided in the supplement as related to the aircraft comparisons.

ln[24] *Why the US?*

The large number of independent measurements available for 2014/2015, especially for the lower troposphere from multiple aircraft campaigns, provides a unique opportunity to be able to investigate and quantify the seasonal variability over a significant area. Moreover, we wanted to focus on a corresponding NOy analysis for which aircraft measurements are a unique source. No such measurements are available with such a frequency for either Europe or China, which would have significantly hampered the analysis. Whilst CAMS provides a regional forecast for air quality using an ensemble of regional models for Europe, for other regions over the globe CAMS relies on global model simulations to provide such information due to the size of the domains. Therefore, we argue that quantification of the model performance in terms of air quality aspects for another region than Europe, with specific emphasis on uncertainty due to atmospheric chemistry modeling, is of interest to the CAMS consortium and wider community. This is the first time that such an in-depth analysis of the CAMS system adopting independent chemical schemes has been performed for the US.

ln [55]: We have now corrected the sentence as : "Note that besides the global system, there is also an operational suite of regional-scale models for providing timely AQF for Europe (Marécal et al., 2015)."

ln [62] : *Is this then a companion paper to how these updates performed globally in the background regions? Why not Europe and Asia too which are heavily polluted too?.*

For assessing model performance in the pristine background away from strong emission sources the referee is referred to the evaluation presented in Huijnen et al. (2019), which pertains to this current study. The reviewer

correctly addresses the need for assessments of model performance for other pollution regions, as well as for background conditions. Analysis of model performance over other polluted regions is expected to take place in future studies. For the European domain, this is less relevant to the user community as the CAMS regional system is more appropriate. For other regions of the world, any analysis and evaluation is somewhat limited by the availability and subsequent interpretation of observational data for evaluation. We have now added the following sentence to the outlook: "Evaluation of model performance for other important polluted world regions, such as Europe and East Asia, will be the focus of future studies."

ln [70]: We have now rewritten the sentence as : "while stratospheric ozone can be well constrained by a linear model combined with ($O_3$) data assimilation"

ln [96]: Apostrophe removed : Volatile Organic Compounds (VOCs)

ln [101-102]: Removed double definition for $O_3$/CO and placed reference within brackets.

ln [154] We have now changed the sentence to "… for organic species by defining a single separate generic tracer species."

ln[159] Reference now corrected.

ln{169] Font in Table changed.

ln[215] Double definition corrected.

ln[236] Font corrected.

ln[264]  'NC' means Not Considered, where a definition has now been included in the table heading.

ln [257] *is the goal to be similar to CBA?*

Although all chemistry modules were expected to include parameterizations for heterogeneous scavenging on aquated aerosol, the implementation choices are left to the individual modeling teams depending on their insight. The teams developing the IFS(MOCAGE) and IFS(MOZART) chemistry versions have so far chosen to adopt the settings as developed for IFS(CBA) , although the IFS(MOZART) configuration does not account for any update on ice and cloud droplets. We now change the text to: "...fully updated. It was chosen to make this parameterization fully consistent with the IFS(CBA) configuration).

ln [272] capitals removed

ln[276] Paragraph break removed

ln[285] model top?

Model top is situated at 1 Pa, please see https://www.ecmwf.int/en/forecasts/documentation-and-support/137-model-levels (last access 29th December, 2021).

ln[286] *Are there still realistic biomass burning that are satellite based?  Is there meteorological data assimilation or is the model nudge to analyzed meteorology?*

No data assimilation is employed in the current model experiments, neither from composition nor from meteorology. The system is initialized daily at 0 UTC with meteorology from ECMWF operational analyses.

*ln[286] what does this mean? Is it a free running model each day?  Does the model only do chemistry every 30 minutes?  How long does it take to run these models for a year like this and how many resources does it use?  Is it cheap or computationally expensive to initialize the model every day?  That might be beyond the scope of the paper or what the authors need to share but I am curious.*

The model is a free running model which uses a 30 minute time step for the chemistry calculations. The reason for choosing this configuration of daily forecasts is because this is how the operational system is configured and applied in most configurations. It is technically possible to run longer, (monthly) experiments using nudged meteorology, which helps to reduce the expense of any simulation but offers less flexibility in terms of system configuration, and is less supported.

ln[289] *why this period? Any large driving meteorological patterns to consider in 2014/2015 like ENSO?*

As we have discussed above, we chose the years 2014/2015 as based on the large availability of measurements within the US, rather than based on a large-scale phenomenon such as ENSO.

ln[307] Double definition removed

ln [312] Now corrected.

ln[314] We have now clarified our sentence to "The variability in the emission estimates compared to previous studies are due to both the use of different data in the derivation of fluxes and trends in the annual emission estimates (2014 versus 2019).

ln [317] Acronyms now added.

ln[325] Figure 1 is now referenced in the text associated with describing the observations.

ln[340] Figure 1 has been modified thus: Stars are used for the locations of the ozonesondes and triangles added for the GFDL surface sites. All Canadian surface sites have been removed from the Figure. The regions defined for the aircraft measurements have been renamed.

ln[377] We now correct to ".. in order to avoid .."

ln(381) We replace the abbreviation EC with an abbreviation representing the various states covered in the measurement campaign, as suggested by Dr. Knowland

ln[395] *Firstly, isn't this manuscript supposed to assess trace gas concentrations of $O_3$, CO and $NO_2$ because the IFS produces Air Quality Forecasts. Why not show the urban and suburban too?*

It is well known that data from suburban and urban sites include roadside measurement stations which invoke short scale gradients which cannot be captured by a global model, even when run at a high resolution. Here small scale meteorology is affected by buildings, concrete and traffic flow, which is why specific small scale models for cities are employed. The chemical emissions are likely to be strongly affected by traffic speed and density, which is not accounted for in the model. Therefore, we choose rural stations only to address this.

*Secondly, the reason the US was chosen is because most of the model evaluation is done in "pristine environments". How does selecting only the clean background sites line up with this reason?*

We only use the word pristine in association with previous studies. In addition to the points discussed above, the reason the US was chosen was due to the current provision of forecasts to existing users and the wealth of independent measurement data available for the year and region for evaluation. The environments sampled here are not considered pristine, which are typically located hundreds of kilometers away from strong sources. For the rural stations this is limited to tens of kilometers.

ln [400] *'how do you know this to be true? No reference to anyone doing this before'*

The process we describe is for removing zeros or values which persist for hours which cannot be true. We feel that data screening is an important step towards conducting a valid comparison. The technique is part of the established 'evaltools' software package which is readily available on the web via https://opensource.umr-cnrm.fr/projects/evaltools (last access 29th December 2021) and refer the referee to the description and information available online.

We add an additional sentence: " We use version v1.0.3 of the evaltools statistical package available online (https://opensource.umr-cnrm.fr/projects/evaltools)."

ln [404] *Comment on use of the year 2014/2015 for surface analysis.*

Our aim is to show a complete annual cycle therefore we limit our comparisons to Jan through to Dec 2014. Moreover, our simulations only ran through till June 2015, thus we do not have a complete simulation for 2015. We add an explanation in the text : "For brevity we do not show any comparisons for 2014".

ln[406] We have renamed the regions in Table 9 as suggested by Dr. Knowland.

ln[436] We have switched the order of JJA and DJF.

ln[438] *How does the PBL factor into this calculation. How many levels are likely included below 1km?*

The choice of 1km is to place the focus on the surface distributions rather than the polluted boundary layer. Increasing the height over which averages are calculated dampens the spatial variability due to mixing in the lower troposphere, therefore not directly relevant to showing a relevant surface distribution. The 137 vertical grid employed means averages are for 20 levels (please see https://www.ecmwf.int/en/forecasts/documentation-and-support/137-model-levels; last access 03-01-2022).

ln[442] .. NOx and VOC emissions. Changed Colorado to Kansas/Nebraska

ln[446] The statement ".. due to reduced transport" has been removed.

ln [451] Remove second definition of OH.

ln[453] Replaced with "ozonesonde"

ln[455] On the request of Dr. Knowland we have changed the sentence to : "For JJA there are positive biases near the surface for Yarmouth (east coast), whereas for Trinidad Head ( west coast) biases are smaller therefore the agreement is more favorable."

ln[456] Thank you for highlighting, we have now changed Texas to Alabama

ln[465] We feel that the different ranges in mixing ratio used for each season provides more information with respect to the divergence across simulations.

ln[466] We now change the figure legend to "situated in the US/Canada"

ln[471] We now move the definition of station acronyms to the main text and out of the figure legend.

ln[481] ' Topography' has been included.

ln[485] *why do you think this was good and others less so?*

We no longer make direct comparisons against CAMSRA due to the lack of assimilation in the other simulations and the use of a different cycle of the IFS which introduce differences not due to the chemical mechanism. Sentence has been rewritten to state that all simulations capture the observations well.

ln[486] Sentence rewritten to state that the station is at an elevated site.

ln[490] The tick marks between months in this figure have been removed and the comparisons shifted such that each point occurs in the middle of a month

ln[496] Corrected table number

ln[514] *Did I miss something, I thought the urban sites were removed and only rural background stations were used? Where should I find this?*

We now remove all discussion related to scenarios which are not shown and refer Dr. Knowland to our response above as to why only rural background stations were selected.

ln[565] Second definition removed

ln[571] Included Eastern Coast in text

ln[572] For clarity we have rewritten the sentence to "The distribution of CO for this season in IFS(CBA) shows signatures of increased pollution visible over large urban centers in e.g. California, Washington state and New York state, in line with figures presented for LT $O_3$ related to regional $NO_x$ emissions (See next section)."

ln[580] We now add a reference with information of biomass burning activity for 2014, where large fires occurred during summertime in Southern Canada near the observational site LEF.

ln[582] *June spikes low in range with obs and then the second half of the year seems positive again. Seems odd, do the authors have any ideas on this?*

Given that we sample the model to take account of the elevated station height (see revised text), thus limiting the influence of regional surface emissions from Denver (see response to Referee #2), we think that variability is governed by long-range transport and consecutive uplift of emissions as seen by other investigators (McDuffie et al., 2016).

ln[590] The x-axis of this figure has been changed to show months of the year

ln[600] the use of brackets to avoid verbose sentences is commonly used in the literature therefore we disagree that we overuse brackets as we maintain the naming culture used throughout the rest of the text.

ln[606] We now remove reference to urban comparisons of the AirNow data.

ln[613] We use 50hPa to bin the measurements resulting in 7-10 independent values depending on the pressure range per campaign. We have included details in the Section 3.2

ln[615] The metric for defining whether vertical profiles are captured well is both the gradients and the inflection in the gradient in the lower atmosphere representing the limit between the boundary layer and the free troposphere. We now change the text to : ".. the shape of the vertical profiles is captured well (i.e gradients and the inflection between the boundary layer and free troposphere) with  .."

ln[617] A reference has been added related to the fracking emissions.

ln[629] Sentence corrected.

ln[630] *This reads as an entire paragraph about figures in the supplemental*

We have now moved the aircraft comparison of $CH_2O$ out of the supplement and into the manuscript itself and subsequently renumbered the figures.

ln[634] Brackets removed.

ln[683] Changed definition of ratio to Capital F

ln[686] The stations are at similar locations for some

ln[704] *Are urban comparisons provided elsewhere?*

We now remove reference to urban comparisons of the AirNow data. See responses above related to this issue.

ln[715] We now make a reference to the Supplementary Information.

ln[726] We have now included concentration brackets in the definition of $[NO]/[NO_2]$ and changed the ratio to S.

ln[747] We have now changed Section 5 into a subsection of Section 4 as suggested.

ln[750] Colour legends now added

ln[770] We have now changed the sentence to : For IFS(MOZ) no $N_2O_5$ conversion on cloud particles and ice droplets is assumed, unlike the other chemistry modules.

ln[784] Figure 16 pertains to $[HNO_3]/[NO_x]$ and not$[HNO_3]/[N_2O_5]$, which is indeed Fig. S10.

ln[799] Indent removed. Brackets introduced for the definition of R'

ln[804] Pearson R used

ln[839] and [843] These values are calculated during the production of the deposition figure. For brevity we do not show all values for different regions but give a general description in the text..

ln[855] Reference brackets fixed

ln[867] Concentration brackets are introduced in the definition of the ratio.

ln[871] now corrected from 0.2 to 2.

ln[876] *So pressure levels between 1000 and 850 hPa?*

The height of the boundary layer is determined by both time of day and underlying terrain, therefore we change the sentence to read "... good agreement in F for the lower atmosphere."

ln[879] ND acronym now included.

ln[890] changed to " in the ECMWF IFS global model". Capitals removed.

ln[902] now changed to " good agreement for the Pacific NorthWest and an overestimation towards the Eastern Coast." using the new regional definitions suggested above.

ln[904] US removed from text

ln[934] Ratios now declared as $[HNO_3]/[NO_x]$

ln[954] $[PAN]/[NO_2]$ now included

ln[961] Comma added.

ln[990] We know replace with 'ozonesonde'.

ln[994] Thank you for highlighting this omission. We now acknowledge NASA for the DISCOVER-AQ campaign.

Refs:

V. Huijnen, P. Le Sager, M Köhler, G. Carver, S. Rémy, J. Flemming, S. Chabrillat, T .van Noije. OpenIFS/AC Atmospheric chemistry and aerosol in OpenIFS cycle 43r3. In preparation for Geosci. Model. Dev., 2022.

McDuffie, E. E., Edwards. P. M., Gilman, J. B., Lerner, B. M., Dubé, W. P., Trainer, M., Wolfe, D. E., Angevine, W. M., de Gouw, J., Williams, E. J., Tevlin, A. G., Murphy, J. G., Fischer, E. V., McKeen, S., Thomas B. Ryerson, T. B., Peischl, J., John S. Holloway, J. S., Aikin, K., Langford, A. O., Senff, C. J., Alvarez II, R. J., Hall, S. R., Ullmann, K., Lantz, K. O. and Brown, S. S.: Influence of oil and gas emissions on summertime ozone in the Colorado Northern Front Range, J. Geophys. Res. Atmos.,121,8712–8729, doi:10.1002/2016JD025265, 2016.

---

## Author Comment (AC2)

Response to Anonymous Referee #2 We thank the referee for the review of our manuscript and in making insightful comments which we feel improve on the final version of the manuscript. We provide detailed responses to the point raised below.

General Comments:

*I was expecting that the authors will evaluate the impact of the updates to the homogeneous and heterogeneous chemistry against the older model version without these updates to demonstrate how important these updates here. However, such a comparison is not presented. Thus, the real value of these updates is not quantified.*

The updates to reaction data and chemical mechanism presented in Tables 1 through 6 were implemented and tested in a stepwise fashion, with these steps being documented in publicly accessible deliverable reports written and submitted to the CAMS consortium. For brevity, and to also address the criticism from Dr. Knowland on the length of the paper, we felt the need to limit the number of simulations to those chosen. Rather than investigating the improvement of the CB05BASCOE scheme, the focus is primarily on the divergence seen across participating members of a proposed chemical ensemble initiative and to focus on seasonal differences and the main reasons for such differences when employed under identical conditions. We now also move the tables related to some of the detailed model revisions to the Appendix, as suggested by the referee.

*The authors compared the three chemistry configurations against the CAMS reanalysis (CAMSRA). At lines 76-77, the authors state that any changes in the operational system should ensure that the change improves the model performance with respect to observations. In most cases (e.g., Figures 4- 6), the performance of CAMSRA is better than all the three individual configurations. Since any of the new configurations here do not outperform CAMSRA, does it mean that these configurations may not be suitable for operational adaptation? Furthermore, the assimilation of several species in CAMSRA makes a direct comparison of three chemistry simulations with CAMSRA difficult.*

The intention of including the CAMSRA dataset was to provide a clear point of reference to a well-known reanalysis which is accessible to the community. We feel it provides some assessment with respect to the three chemical schemes employed in a more updated cycle of the IFS run at a higher horizontal resolution. Our aim was not to document each model improvement explicitly, but we agree with the reviewer confusion may have been raised throughout the discussion of model evaluations on this aspect. Clearly the current model performance is found occasionally to be of poorer quality as CAMSRA, but, as the reviewer also notes, this is at least in part caused by actual data assimilation, particularly for trace gases that are well-constrained (CO). Updates made to the IFS cycles are separate from updates made to the chemistry modeling aspect. A cycle update also changes to the data-assimilation configuration, the meteorological model. and emissions estimates that are applied. A rigorous assessment of the net changes is made and documented publicly in various system upgrade reports hosted at https://atmosphere.copernicus.eu/eqa-reports-global-services (last access: 21-12-21). We now clarify better the scope of our assessment by adding the following sentence: "However, attributing the differences in performance of chemical updates without data assimilation against CAMSRA is complicated by the simultaneous changes in IFS cycle and emission estimates since CAMSRA was produced."

*The choice of US as a polluted region over Asia is not motivating enough given that anthropogenic emissions in Asia are much higher than the US.*

The large number of independent measurements available for 2014/2015, especially for the lower troposphere from multiple aircraft campaigns, provides a unique opportunity to be able to investigate and quantify the seasonal variability over a significant area. Moreover, we wanted to focus on a corresponding NOy analysis for which aircraft measurements are a unique source. No such measurements are available with such a frequency for either Europe or China, which would have significantly hampered the analysis. Whilst CAMS provides a regional forecast for air quality using an ensemble of regional models for Europe, for other regions over the globe CAMS relies on global model simulations to provide such information due to the size of the domains. Therefore, we argue that quantification of the model performance in terms of air quality aspects for another region than Europe, with specific emphasis on uncertainty due to atmospheric chemistry modeling, is of interest to the CAMS consortium and wider community. This is the first time that such an in-depth analysis of the CAMS system adopting independent chemical schemes has been performed for the US.

*57-58: Operational air quality forecasts over the US are not only provided by the CAMS system. The US has their National Air Quality Forecasting Capability (NAQFC) that provides air quality forecasts at 12 km grid spacing twice per day and the NAQFC forecasts are used by decision- makers to issue advance notices and warnings for air pollution episodes. Since lines 43-55 discuss aspects and benefits of air quality forecasting, it is relevant to discuss NAQFC here.*

We agree that we should prevent confusion as to the status of the CAMS system to provide operational air quality forecasts on a global scale. Therefore, we have added the following text related to other forecasting systems:

"Forecasting services for the US are also provided by the National Air Quality Forecasting Capability (NAQFC,) for ozone ($O_3$), dust and smoke (e.g Chai et al., 2013; Lee et al., 2017), which utilizes the CMAQ model (Byun and Schere, 2006) driven by WRF-NMM meteorological model (e.g. Eder et al, 2009), provided at a 12 x 12km resolution. The model is regional meaning that prescription of boundary conditions is necessary to account for effects of long-range transport. One other notable system for AQF is the NASA GEOS Composition Forecast Modeling System (GEOS-CF v1.0; Keller et al, 2021), which is a global system similar to CAMS using the GEOS-chem chemical module coupled to the GEOS atmospheric model (v5) run at a global resolution of 25 x 25km."

*Lines 63-64: Anthropogenic emissions in the US are much lower than Asia and thus it is hard to see this as a motivation for focusing on the US.*

Please see our response to the second major point above.

*Line 99: To what heterogeneous reactions you are referring here?*

Principally the conversion of $N_2O_5$ to $HNO_3$, which has been improved on in the chemical schemes compared to previous studies and is considered critical for controlling the NOx regeneration cycle. We change the sentence to: "Huijnen et al. (2019) has highlighted the importance of heterogeneous reactions, especially the conversion of $N_2O_5$ to $HNO_3$, to explain differences".

*Line 105: I suggest replacing the term "mini-ensemble" with "multiple chemistry simulations".*

The reviewer is correct that we need to be careful with any referencing of our model configuration that is based on three model versions with independent chemistry as a 'mini-ensemble', because an actual ensemble should consist of a much more independent, well-chosen selection of members, and also requires assessment of ensemble statistics. The use of the referencing of three model versions as 'mini-ensemble' is essentially chosen from a practical perspective, in combination with an implicit future perspective where these chemistry versions could form a building block of constructing an actual global chemistry ensemble. We now update the sentence and write: "Furthermore, by applying all three model versions **with largely independent chemistry modules, for brevity here referred to as a chemistry mini-ensemble,** uncertainty ranges due to inaccuracies in the chemical component of the forecast can be estimated"

*Line 144: Do all the chemistry configurations use the same quantum yield and cross sections for calculation of photolysis frequencies? Also, how are the effect of aerosols treated in photolysis rate calculation? If not, can the 5% differences in photolysis frequencies be attributed to these factors?*

There is no homogenization across chemistry configurations regarding the calculation of photolysis frequencies, where differences have already been described in the previous literature. The Aerosol Optical Depth is used to attenuate the photolytic flux within each approach. We have added the sentence: "Different photolytic data is used for each of the chemistry configurations."

*Tables 2-5: Please include a description of various symbols (Ko, R, T, and QY etc.) used in the rate expression column in the captions of these table.*

We have added the following definition in the Table heading of Table 2: Key to abbreviations: T – Temperature (°K), K0 - Low Pressure Limiting Rate Constant, $K_{inf}$ - High Pressure Limiting Rate Constant, QY – Quantum Yield and R – product yield.

*Line 168: Do you provide HONO emissions to the model as well?*

No HONO emissions are introduced directly. We now state this in the header of Table 7 addressing the emissions applied in the simulations.

*Tables 3-5: Since you are not validating the updated isoprene, aromatic, HCN, and CH₃CN chemistry, I think Tables 3-5 can be moved to supplementary material.*

We thank the reviewer with this suggestion and will move some of the Tables associated with the model updates not attributed to Nitrogen Oxides.

*Line 442-443: The minimum is not centered around Colorado but rather is spread over Oklahoma, Nebraska, South Dakota, North Dakota, and parts of Texas.*

We have now re-written the sentence to: "There is a minimum in Central rural regions around Kansas/Nebraska and North/South Dakota, with a variability of between 35-45 ppb in IFS(CBA)."

*Line 446: Can you explain why the decrease in MOZ is due to reduced transport? What are the source regions affecting the surrounding oceans?*

This is likely due to the lower PAN formation in IFS(MOZ) which reduces the fraction of $NO_x$ (c.f. Figure 18 and Figure S7) which can be transported away from the main source regions, which alters $O_3$ formation

We have modified the sentence accordingly: "Over the surrounding oceans, IFS(MOZ) has a decrease in mixing ratios of between 5-10 ppb due to reduced transport of pre-cursors such as PAN (see $NO_y$ discussion below)."

*Fig S1: The differences in OH are interesting. While it makes sense that higher ozone can lead to higher OH levels through photolysis but I am unable to understand why OH increase in summertime in MOZ over the western US despite of a decrease in ozone mixing ratios. Any thoughts?*

We agree that this is a somewhat surprising result which is difficult to explain being a consequence of the different reaction kinetics employed across the chemical schemes, which drive the OH production, recycling, and loss (e.g. Lelieveld et al., ACP, 2016). For example, primary production of OH from $O_3$ photolysis is strongly affected by the photolysis rate employed. Analysing the global budget terms shows that the net loss of $O_3$ via photolysis is around 12% higher in IFS(MOZ) compared with IFS(CBA), which will contribute towards the regional differences shown for the US. The extent of OH recycling is also influenced by the rate of oxidation of NO by $HO_2$, with higher summertime NO in IFS(MOZ) (Figure S2) introducing a greater recycling rate for the chemical scheme. Loss will then be the cumulative scavenging across reactants, where the kinetics of OH + CO exhibit a different pressure dependency without the inclusion of an intermediate in IFS(MOC) as for IFS(CBA) and IFS(MOZ) schemes. Due to the size and expense of the simulations it was not able to provide a full 3D budget analysis of the US region, which would provide insight in the changes in reaction cycles. We now add an additional sentence: "The OH primary production is globally higher in IFS(MOZ) than IFS(CBA), likely contributing to the higher summertime OH in this configuration."

*Figure 4: The observed ozone seasonal cycle at NWR looks suspicious. Most of the sites in Colorado show a summertime maximum. I looked at multi-year hourly ozone time series from NOAA GML website (see below) and found that there are a lot of measurement gaps at NWR during 2014. So, I recommend revisiting the data processing and evaluation at this site and revising the corresponding text accordingly.*

We are somewhat puzzled by this point as when we revisited the original measurement data (hosted at: https://gml.noaa.gov/aftp/data/ozwv/SurfaceOzone/NWR/2014/ ; last access 27th December 2021) we do not find any significant gaps in the measurements for $O_3$ during 2014, with a (changing) value

being given every hour for each day and month allowing a valid observational monthly mean to be assembled. Therefore, we do not make any changes with respect to observational data frequency. We show a graphic of the dataset used below:

[Figure]

The hourly O$_3$ data used for the comparison presented for the NWR site, 2014.

*Lines 535-540: Here you use ppb as ozone units while all the earlier discussion is in micrograms/m3. Please use consistent units.*

We adopt the units provided by the various measurement networks therefore use these for our comparisons, where conversion of micrograms/m3 requires co-located pressure and temperature values associated with the measurements.

*Line 581-582: Can you provide more insight into the different seasonal cycles of CO between the model and observations at NWR?*

The observational site at Niwot Ridge is located approximately 35 km west of Boulder on top of a mountain range at around 3000m altitude. This site acts as an ideal location at which to sample background concentrations of CO, with no anthropogenic emission sources near the site. This means that this site is challenging when sampled by our global model with a resolution of 80 km that includes emission sources of the Denver area. To account for the fact that the measurement site is situated on a mountain, we sample the third (CAMSRA) and fifth (IFS(Chem)) model levels, respectively, to remove artifacts due to the different vertical and horizontal resolutions of the various datasets. In this way we make sure not to confuse the comparisons with signatures of local emissions – which indeed play an important role due to grid size. In future model configurations the model resolution should be increased to better capture more regional effects as encountered here. We now modify the text accordingly: "A signature exists at NWR from the chemical processing of polluted air masses from the Denver region during summertime (McDuffie et al., 2016), where all members show similar positive biases, where sampling accounts for the elevated station height."

[Figure]

Left Panel: Adopting the surface level for interpolation Right Panel: Taking level 5 above surface for IFS and level 3 for CAMSRA.

*Lines 775-783: This is a good discussion but it would have been great to perform a couple of sensitivity experiments using just one of the chemistry configurations to understand the impact of conversion efficiency on $N_2O_5$ hydrolysis.*

This would be a point of interest to further investigate global variability, possibly for different years and locations where observations permit. But funding and resource limits mean that such studies are not currently feasible or planned.

---

## Referee Report (RR1)

Second Review of "*Regional evaluation of the performance of the global CAMS chemical modeling system over the United States (IFS cycle 47r1)*" submitted by Jason E. Williams et al. to Geoscientific Model Development

The authors made great efforts to address my comments and concerns from their initial submission and the manuscript is greatly improved. I have a few technical edits listed below, and once addressed I recommend this paper for publication.

**Comments:**

Line 61: The NAQFC has a comma after it, as if the authors were going to note something else. What I found lacking was the acknowledgement of NOAA with NAQFC and recommend adding it here.

Line 61: There is also a period and comma missing in "(e.g**.,** Chai et al., 2013".

Line 63: I find the wording "The model is regional meaning" awkward and recommend changing this to something like

> "This system is a regional model configuration, therefore, it requires lateral and chemical boundary conditions from a global model to account for effects of long-range transport."

Line 64-67: Thank you for the acknowledgement of the NASA GEOS-CF system. I suggest changing "One other" to "Another notable system" since there are other global models (e.g., the Finish SILAM, NCAR's WACCM) and regional models (e.g., ECCC, NCAR's WRF-Chem and NOAA's RAP-Chem) which also forecast over the US. I'd hesitate to say that GEOS-CF is "similar to CAMS" unless you say specifically that they are both CCMMs (coupled chemistry and meteorology models) as here the GEOS-CF uses GEOS instead of the IFS and GEOS-Chem is not an option in the IFS as far as I am aware. Note there is a missing period after et al. in the Keller et al. reference. Also, we are no longer referring to GEOS by its version number so I recommend removing the (v5). Altogether, I suggest changing this sentence to something like:

> "Another notable system for AQF is the NASA GEOS Composition Forecast system (GEOS-CF v1.0; Keller et al**.**, 2021); it is a global coupled chemistry and meteorology model like CAMS but uses the GEOS atmospheric circulation model coupled to the GEOS-Chem chemistry module and run at the global resolution of 0.25° x 0.25° on 72 vertical layers."

Line 87: It is likely fine, but the radicals OH and HO2 were not defined at first use (hydroxyl radical and hydroperoxyl radical). I only note it here because in line 111 the authors define the "hydroxyl radical (OH)" so it should likely be done here at first use.

Line 166-168: I still find this sentence awkward. I think it is the "is computed" that is throwing me off. I suggest something like this:

"In the MBA, the radiative transfer calculation is performed with a two-stream solver using the absorption and scattering components introduced by gases, aerosols and clouds which are computed online for each of the predefined band intervals."

Line 240: The SOG was defined already at line 214 (and SOA defined earlier on Line 181). Is there a reason the authors have redefined it here?

Line 270: In response to one of my comments on the initial draft, the authors changed BENZENE to Benzene, but TOLUNE and XYLENE have been reduced simply to TOL and XYL with no explanation, instead of using lower case letters. This current version is confusing as "TOL" in the line above refers to a lumped aromatic. Can these also be changed to tolune and xylene?

Line 309: Add a comma after "Huijnen et al. (2019),"

Line 325: State in this first paragraph that Yarmouth is in Canada and given its close proximity to the US it is a good proxy for New England and that is why you have included it in your study of US air quality.

Line 330, Figure 1 caption: GMDL is not defined in the paper. In line 340, "NOAA ESRL Global Monitoring Laboratory" is written. This needs to be better defined in the main text and matched in the figure caption. Also, was it a choice not to add dots for the AirNow CO stations that would match with the Table 5 count of stations?

Line 338-342: The three sites THD, BAO and NWR used for O3 are part of the NOAA ESRL GML, so the Niwot Ridge does not need to be redefined in line 342 unless it really is at a different longitude. Also, it may be good to move up the definition of BAO and THD from line 472-473 here to be closer to the Figure 1 where we see they are used to label the ozonesonde locations.

Line 380: I think you caught most of the region updates throughout but double check as here I see the use of "Col" instead of "COL" and "TX" instead of "SC" as given in Table 4.

Line 400, Table 5: Some of the regional domains still included latitude or longitude bands that would include Canada. Particularly, in New England, 65 °W is further east than necessary if US only, as this box should mark out the region for AirNow observations and does not need to include Yarmouth, Nova Scotia. For North/South Dakota and Pacific Northwest, both go up to 50 °N in the table but the US border stops at 49° and the boxes in Figure 1 were reduced to reflect that. What concerns me is the number of total stations has not changed, which leads me to wonder if the analysis included any of the Canadian stations in the initial submission or not. This should be confirmed so the authors are confident that only the US stations are included in the analysis in the final version. There were only a few that were likely included in the Pacific NorthWest, but it looks like half the O3 red dots that were once in the New England region were removed when the Canadian sites were removed.

Line 456-457:  Here the authors refer to ozonesondes composites "located across the US" which is not correct with Yarmouth in Canada.  This is why there needs to be a statement at Line 325 that Yarmouth is being used as a proxy for New England so that it is appropriate for the authors to write "Yarmouth (east coast)". I do appreciate that Figure 3 caption was updated to have "across the US/Canada".

Line 466, Figure 3:  The authors could add here "Note, x-axis and y-axis scales vary." or something to that effect.

Line 491, Figure 4:  The legend was removed from the left panel and should be added in as per the figure caption.  However, the THD panel is very busy, so I recommend adding it back in to the bottom left corner of the NWR (middle) panel.

Line 508:  I recommend changing this to:
   "During the first few months of 2014, a significant mean positive bias exists for more northerly regions across the chemical modules, with IFS(CBA) biases up to 20 ug m$^{-3}$ and IFS(MOZ) biases as high as 40-60 ug m$^{-3}$."

Line 553:  VOC should be plural: VOCs

Line 687:  I made a similar comment in the Conclusion section of the initial draft which was addressed.  Here "Western US" refers to a portion of the USA while "Eastern US" is one of your defined regions.  It would likely help to change Western to lowercase "western US" and could add in brackets the regions from Table 8 that the authors associate with the lower biases (e.g., Colorado, California/Nevada).

Line 715:  This is a sentence where you could add the brackets to help with brevity
   "Low (high) values for S are indicative of an equilibrium which favors O3 production (destruction)."
I do like this way of writing.  In the earlier cases, I found the double parentheses with the model names hard to read.

Line 808, Figure 18:  There is one site in Canada.  Similar to Figure 3 caption, I suggest this is changed to "across the US/Canada for"

Line 890-938:  I mentioned this on the first draft, it took me by surprise that the full names of the chemical modules were written out instead of the short names used throughout the paper. It may be a conscious choice, but I remember finding this made the conclusion feel separate from the rest of the paper.

Line 974:  ozone-sonde still needs the hyphen removed.

---

## Author Response (AR2)

We thank the reviewer for her review of our corrected manuscript. The response to technical corrections suggested by Dr. Knowland is given below:

1. Ln 61: NOAA accreditation has now been added.
2. Ln 61: Corrected
3. Ln 63: The sentence suggested has been included.
4. Ln 64-67: Changes have been made to the structure of the sentence as suggested.
5. Ln 87: Both radicals are now defined, and the second definition for OH removed.
6. Ln 166-168: The sentence suggested has been included.
7. Ln 240: Second definition of SOG now removed.
8. Ln 270: The aromatic species have now been defined as requested.
9. Ln 309: A comma has been added.
10. Ln 325: We have changed the text to include the specifics regarding the Yarmouth station.
11. Ln 330: GFDL has now been defined in the text. We find that adding the CO stations on the figure would not aid clarity for the other stations.
12. Ln 338-442: We use the first definition for the NWR station in the proceeding discussion of measurement stations. We think that the paragraph should remain as is as it is near the figure in the final manuscript.
13. Ln 380: Acronyms now corrected.
14. Ln 400: Thank you for raising this oversite in the tables, where the maximum latitude is now declared at 49N. The number of stations is output in the interpolation code rather than from this figure which simply reads in station locations. We ensure that Canadian stations are not included by defining the station type as 'unknown' in the listings file thus skipped in all analysis. Thus, the number of stations has not changed even though we removed stations from the figure as requested.
15. Ln 456-457: We now add US/Canada in the text to address this point.
16. Ln 466: We have added: "Please note that both the x- and y-axis vary with respect to station.".
17. Ln 491: We have defined the color scheme in the figure legend to avoid cluttering the figure panels.
18. Ln 508: We have now reformulated the sentence as requested.
19. Ln 687: we now change the text to: "with biases towards the west of the US being lower than the East of the US"
20. Ln 715: For the aims of consistency and considering previous comments we do not change this sentence.
21. Ln 808: Us/Canada added.
22. Ln 880-938: The conclusions and abstract sections are typically the go-to sections of any paper for a reader who just wants a summary of the findings therefore we have no problem with declaring each model version using the full nomenclature.
23. Ln